# A GRAPH NEURAL NETWORK APPROACH TO AUTO-MATED MODEL BUILDING IN CRYO-EM MAPS

**Kiarash Jamali, Dari Kimanius, & Sjors H.W. Scheres**
MRC Laboratory of Molecular Biology
Cambridge, UK
`{kjamali,dari,scheres}@mrc-lmb.cam.ac.uk`

## ABSTRACT

Electron cryo-microscopy (cryo-EM) produces three-dimensional (3D) maps of the electrostatic potential of biological macromolecules, including proteins. Along with knowledge about the imaged molecules, cryo-EM maps allow *de novo* atomic modeling, which is typically done through a laborious manual process. Taking inspiration from recent advances in machine learning applications to protein structure prediction, we propose a graph neural network (GNN) approach for the automated model building of proteins in cryo-EM maps. The GNN acts on a graph with nodes assigned to individual amino acids and edges representing the protein chain. Combining information from the voxel-based cryo-EM data, the amino acid sequence data, and prior knowledge about protein geometries, the GNN refines the geometry of the protein chain and classifies the amino acids for each of its nodes. Application to 28 test cases shows that our approach outperforms the state-of-the-art and approximates manual building for cryo-EM maps with resolutions better than 3.5 Å [1].

## 1 INTRODUCTION

Following rapid developments in microscopy hardware and image processing software, cryo-EM structure determination of biological macromolecules is now possible to atomic resolution for favourable samples (Nakane et al., 2020; Yip et al., 2020). For many other samples, such as large multi-component complexes and membrane proteins, resolutions around 3 Å are typical (Cheng, 2018). Transmission electron microscopy images are taken of many copies of the same molecules, which are frozen in a thin layer of vitreous ice. Dedicated software, like RELION (Scheres, 2012) or cryoSPARC (Punjani et al., 2017), implement iterative optimization algorithms to retrieve the orientation of each molecule and perform 3D reconstruction to obtain a voxel-based map of the underlying molecular structure.

Provided the cryo-EM map is of sufficient resolution, it is interpreted in terms of an atomic model of the corresponding molecules. Many samples contain only proteins; other samples also contain other biological molecules, like lipids or nucleic acids. Proteins are linear chains of amino acids or residues. There are twenty different canonical amino acids that make up proteins. All of these amino acids have four heavy (non-hydrogen) atoms that make up the protein's main chain. The different amino acids have different numbers, types, and geometrical arrangements of their side-chain atoms. The smallest amino acid, glycine, has no heavy side chain atoms; the largest amino acid, tryptophan, has ten heavy side chain atoms. Typical proteins range in size from tens to more than a thousand residues. Typically, the electron microscopist knows which protein sequences are present in the sample. The task at hand is to build the atomic model, which identifies the positions of all atoms for all proteins that are present in the cryo-EM map. For each residue, there are two rotational degrees of freedom in the conformation of its main chain, the $\Phi$ and $\Psi$ angles. Distinct orientations of the side chains provide additional conformational possibilities, the number of which depends on the type of amino acid (figure 1).

---

[1] Code and weights are open-source and can be accessed at `https://github.com/3dem/model-angelo`

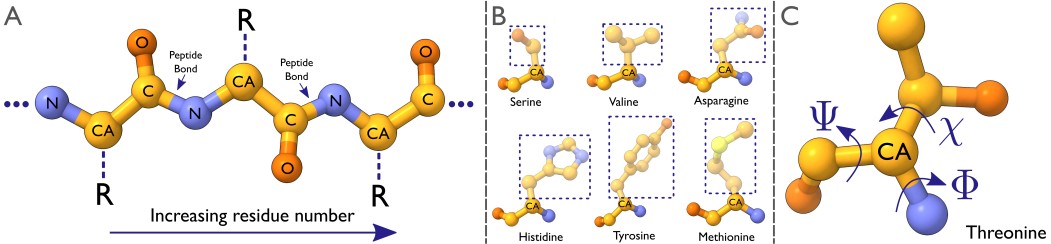

Figure 1: **(A)** shows a peptide backbone where the chain is ordered from left to right. Arrows mark the peptide bonds between the C atom of one residue and the N atom of the next. **(B)** shows six amino acids, pointing out the differences in the side chains, marked by the outline. **(C)** shows the Φ and Ψ angles of the backbone and the additional rotatable bond of the side chain for threonine.

Atomic model building in cryo-EM maps is typically done manually using 3D visualisation software, (e.g. Emsley et al., 2010; Pettersen et al., 2021), followed by refinement procedures that optimize the fit of the models in the map, (e.g. Murshudov et al., 2011; Croll, 2018; Liebschner et al., 2019). Often, in areas of weak density in the map, one cannot discern the amino acid identity of residues from the map alone and sequence information has to be used to make an accurate assessment. Manually building a reliable atomic model *de novo* in the reconstructed cryo-EM map is considered to be difficult for maps with resolutions worse than 4 Å. Although the task is more straightforward for maps with resolutions better than 3 Å, it still typically requires large amounts of time and a high level of expertise.

Machine learning has recently achieved a major step forward in structure prediction for individual proteins (Jumper et al., 2021; Baek et al., 2021). In these approaches, the sequence information of proteins and their evolutionary related homologues is used to predict their atomic structure without the use of experimental data. In addition, protein language models, which are trained in an unsupervised fashion on the amino acid sequences of many proteins, have also provided useful results in protein structure prediction (Lin et al., 2022; Wu et al., 2022a). Although these techniques are not yet capable of reliably predicting structures of the larger complexes that are typically studied by cryo-EM, their success for individual proteins inspired us to explore similar approaches for automated model building in cryo-EM maps.

In this paper, we present a single integrated GNN that combines the voxel-based information from the cryo-EM map with information from the protein sequence through a protein language model, and information from the topology of the graph through invariant point attention (IPA) (Jumper et al., 2021). For 28 test cases, we demonstrate that our approach approximates the accuracy of manual model building for maps with resolutions better than 3.5 Å.

## 2  PRIOR WORK

**Automated model building**. Automated approaches for atomic modeling in the related experimental technique of X-ray crystallography have existed for many years (for example, Perrakis et al., 1999; Cowtan, 2006; Terwilliger et al., 2008). Some of these approaches have also been applied to cryo-EM maps. For example, the PHENIX package builds models that are on average 47% complete for cryo-EM maps with resolutions worse than 3 Å (Terwilliger et al., 2018). For similar maps, MAINMAST, an approach that was designed to build $C^\alpha$ main-chain traces in cryo-EM maps, often produces models with root mean squared deviations (RMSDs) in the range of tens of Å (Terashi & Kihara, 2018). Relatively incomplete models, with large residuals, have limited the impact of these techniques on automated model building in the cryo-EM field thus far.

More recently, Deeptracer (Pfab et al., 2021), the first deep-learning approach for automated atomic modeling in cryo-EM maps, was reported to outperform these earlier approaches. Deeptracer uses U-Nets (Ronneberger et al., 2015) to construct an atomic model *de novo* in the cryo-EM map. In contrast to our work, Deeptracer does not integrate the sequence information with the U-Net, and it does not use a graph representation of the protein chain during model refinement. Instead, Deeptracer treats the entire problem as a segmentation and classification problem. Thereby, it also does

not have support for refining already built models or performing multiple recycling steps. Although Deeptracer predicts amino acid types for each residue, it only builds atoms for the main chains.

There have also been reports to dock and morph the output from protein structure prediction programs, like AlphaFold2 (Jumper et al., 2021), to fit cryo-EM maps (He et al., 2022; Terwilliger et al., 2022). Such approaches suffer when proteins change their conformation in complex with others and are likely to propagate errors in the structure prediction. Thus, it seems sensible to design a neural network approach that integrates both the cryo-EM map and the sequence and protein structure primitives to produce a more reliable structure. This is the essence of our approach.

**GNNs for proteins**. A number of different approaches to modeling proteins with GNNs have been proposed recently. This includes modeling the protein with torsion angles (Jing et al., 2022), SE(3) equivariant graph neural networks (Ganea et al., 2022), and SE(3) invariant GNNs (Dauparas et al., 2022). In our approach, the ordering and connectivity of residues are unknown and have to be inferred, hence representations that require this to be known *a priori*, such as the torsion angle representation, are inappropriate. Furthermore, the relative orientation of the model and the cryo-EM map is important in model building, which makes SE(3) invariant representations ill-suited. Thus, the most natural graph representation is an SE(3) equivariant one. In this project, we choose the backbone frame representation that was first introduced in AlphaFold2, but is now also used in other protein prediction networks (e.g. Wu et al., 2022a; Lin et al., 2022).

## 3 METHODS

### 3.1 GRAPH INITIALIZATION

We start by identifying the positions of the $C^\alpha$ atoms [2] of all residues in the map, which will form the nodes of our graph. This part of the pipeline is formulated as a straightforward segmentation problem, similar to prior work discussed in section 2. That is, the cryo-EM map $V \in \mathbb{R}^N$, where $N$ is the number of voxels, has an associated binary target $T^* \in \{0, 1\}^N$, where 1 represents the existence of a $C^\alpha$ atom in the voxel and 0 the lack of it. Since the minimum distance of two $C^\alpha$ atoms is 3.8 Å, resampling the cryo-EM voxel maps with a pixel size of 1.5 Å ensures that there is no voxel that contains more than one such atom. The goal then becomes to train a neural network $f_\theta(V) \approx T^*$.

#### 3.1.1 NETWORK ARCHITECTURE

We implemented $f_\theta(V)$ as a residual network (He et al., 2016) inspired by the Feature Pyramid Network (Lin et al., 2017a). First, we changed all convolutions to 3D convolutions and changed batch normalization layers (Ioffe & Szegedy, 2015) to instance normalization layers (Ulyanov et al., 2016), as the statistics should not be averaged across instances of a batch since local boxes of cryo-EM density might be from different sections of the map with large differences in scale. No noticeable effects on performance were observed between ReLU and LeakyReLU (Xu et al., 2015), so ReLU was selected due to its improved computational efficiency. We also shifted the network parameters from the low-resolution part of the model to the high-resolution part, and we changed the order of operations so that global information about the structure is more directly integrated into the model at an early stage (for detailed pseudo-code that contains the exact number of parameters and order of operations, see algorithm 1 in A.3.1). In this manner, large-scale structural features, which are recognized at lower resolutions, become easier to integrate with the classification task.

#### 3.1.2 TRAINING DATASET

The dataset starts with 6351 cryo-EM maps with a resolution higher than 4 Å, downloaded from the EMDB (Lawson et al., 2016) on 01/04/2022, and the corresponding PDB files that were downloaded from RCSB PDB (Burley et al., 2021). A portion of these was manually checked for orientation issues, large unmodelled regions, and the existence of large modeled regions that do not correspond to the cryo-EM map. This led to ∼700 manually curated pairs for the first round of training. Then,

---

[2]Atom names in this document follow the PDB (Protein Data Bank) naming convention. See the PDB atomic coordinate and bibliographic entry format description

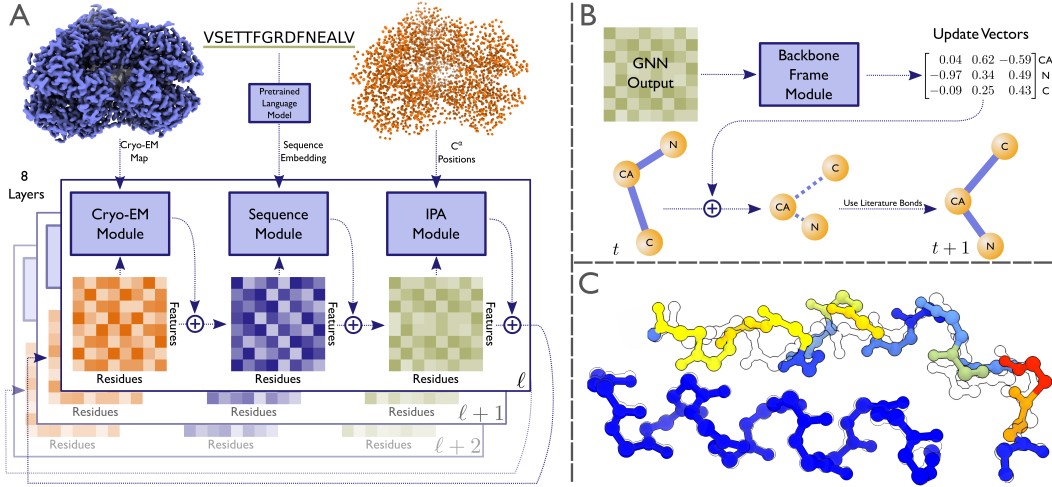

Figure 2: **(A)** shows the schematic of the GNN and how the 8 layers iteratively refine the feature vectors. **(B)** illustrates how the Backbone Frame module updates the positions of the backbone atoms. Finally, **(C)** contains two examples of high confidence (dark blue colour) and low confidence (yellow and red) predicted backbone regions for PDB entry 7Z1M. The confidence measure is a good predictor of fit to the deposited backbone model (shown in outline).

using the first trained model, we were able to automatically detect issues with the rest of the map-structure pairs and prune them to ∼3200 structures. The pruning was based on a cutoff of 70% precision and 70% recall of the model output $C^\alpha$ positions compared to the ground-truth PDB coordinates, at a cutoff of 3 Å. This produced the dataset that was used for training the model.

### 3.1.3 TRAINING

We define the true number of the $C^\alpha$ atoms, or residues, in a map as $M^*$. To ameliorate the issue with class imbalance between the number of $C^\alpha$ atoms and the number of voxels ($M^* \ll N$), we chose focal loss (Lin et al., 2017b) over binary cross entropy loss. Additionally, we up-weighted the loss of voxels with $C^\alpha$ atoms and used a combination of auxiliary loss functions to get an acceptable trade-off between precision and recall. We up-weight the $C^\alpha$ containing voxels in the focal loss with the ratio between the true negatives and true positives, using the following formula:

$$w_x = \frac{N - M^*}{M^*}\chi(x) + (1 - \chi(x)) \tag{1}$$

where $w_x$ is the weight for voxel $x$ and $\chi$ is the characteristic function of whether the voxel $x$ contains a $C^\alpha$ atom. This formula ensures that half of the loss of a map comes from empty voxels and half from the $C^\alpha$ atom containing voxels. We found that also using Tversky loss (Hashemi et al., 2018) in the later parts of training allowed higher recall at the expense of precision. Lastly, to improve generalization to variations in experimental and data acquisition settings, we applied data augmentation schemes to the cryo-EM maps. We added random colored noise to every map and performed random sharpening/dampening by sampling a B-factor (see Rosenthal & Henderson, 2003) from a uniform distribution between -30 Å$^2$ and 30 Å$^2$. Lastly, we applied random rotations that are integer multiples of 90° to both the cryo-EM map and the targets in order to have the model learn different orientations while avoiding interpolation effects.

### 3.2 GRAPH REFINEMENT

Next, we define a GNN $g_\phi$ that is trained to refine the position of all $C^\alpha$ atoms in the graph, to map each of them to an individual residue in the user-provided input sequence, and to provide coordinates for all the atoms in the residues:

$$g_\phi\left(X^{(n-1)}, F^{(n-1)}, V, S\right) = \left(X^{(n)}, F^{(n)}, G^{(n)}, P^{(n)}, O^{(n)}\right) \tag{2}$$

In the above, $X^{(n)} \in \mathbb{R}^{M \times 3}$ are the $C^\alpha$ positions at iteration $n$, $F \in \mathbb{R}^{M \times 3 \times 4}$ are the affine frames defined by the $C^\alpha$, C, and N atoms of the backbone, $G \in \mathbb{R}^{M \times 4 \times 2}$ are the torsion angles of the side chains, $P \in \mathbb{R}^{M \times 20}$ is a probability vector for all twenty amino acids for each residue, and $S \in \mathbb{R}^{M \times 1280}$ are protein sequence embeddings of all residues in the input sequence, and $O \in \mathbb{R}^M$ is a per-residue confidence prediction. The backbone affine frames ($F$) and the side-chain torsion angles ($G$) are similar to what is used in AlphaFold2 (Jumper et al., 2021), see figure 2. Together $(X, F, G, P)$ provide enough information to calculate all-atom coordinates for the proteins in the cryo-EM map.

The overall training objective is to acquire $g_\phi \left( X^{(n)}, F^{(n)}, V, S \right) \approx (X^*, F^*, G^*, P^*)$, where $X^*$ is the set of $C^\alpha$ positions in the training data, $F^*$ and $G^*$ are calculated from the atom coordinates for every residue in the training data, and $P^* \in \{0, 1\}^{M^* \times 20}$ is a one-hot encoding of the amino acid classes in the training data. At iteration $n = 0$, the graph is initialized with $M$ nodes with random $F^{(0)}$, and with $X^{(0)}$ set to the coordinates of those voxels predicted to contain $C^\alpha$ atoms by the graph initialization step in section 3.1.

### 3.2.1 NETWORK ARCHITECTURE

The GNN consists of eight consecutive layers, each of which contains three main modules that are based on the attention algorithm (Bahdanau et al., 2014). Each module extracts information relevant to its modality and updates the feature encoding of the graph nodes with a residual connection, (see figure 2A and algorithm 9 in A.4.2).

The first module is the *Cryo-EM Attention module*, which allows the GNN to look at the density around each $C^\alpha$ atom with convolutional neural networks (CNN), as well as the density linking it to its neighbouring nodes, to update its representation. This is accomplished with a mix of a graph-based attention module and feature extraction using CNNs. The edge features that determine the attention keys are calculated based on CNN embeddings of cuboids (elongated cubes in the direction of the neighbour) extracted from the cryo-EM map spanning each edge that connect a node with each of its neighbours. These regions of the cryo-EM map capture the connectivity between residues, e.g. peptide bond or side-chain:side-chain interactions, and inform the attention scores. The query and value vectors are generated by the feature vector of each node. Additionally, a cube centered at each node is extracted from the cryo-EM density with an orientation defined by the backbone affine frame of each node and passed through a convolutional feature extraction network. Finally, the extracted features from the centered cubes are concatenated with the attention output and projected to create the new feature representation. For precise details about the algorithm, see algorithm 10 in A.4.2.

The second module is the *Sequence Attention module*, where each node searches against the input sequence embedding to find the relevant entries that best fit its features. This is a conventional encoder-only transformer module, similar to that used in Devlin et al. (2018). The user-provided sequence is embedded using a pre-trained protein language model (ESM-1b) (Rives et al., 2021; Lin et al., 2022), which only uses the primary sequence, instead of multiple sequence alignments (MSAs). Generally, MSAs improve the results of protein prediction algorithms by giving them access to co-evolutionary information (Jumper et al., 2021), and they need less learnable parameters than algorithms that rely on a single sequence only (Rao et al., 2021; Lin et al., 2022; Wu et al., 2022b). However, because the cryo-EM map already provides sufficient information about the global fold of the proteins, we chose not to use MSAs, as their calculation would have made using our approach more difficult. For additional details about this module, see algorithm 11 in A.3.2.

The third module is the *Spatial Invariant Point Attention* (IPA) module, which allows the network to update its representation based on the geometry of the nodes in the graph. This module is inspired by the module with the same name in AlphaFold2 (Jumper et al., 2021), although simplifications have been made to better fit the problem at hand. Each node predicts query points based on its current representation in its own local affine frame; these points are transformed into the global affine frame (by application of $F$ to the predicted point); the distance is calculated between each node's query points and its neighbours; and based on the sum of the distances of the query points to the neighbouring nodes, each of these nodes get an attention score that is used to update the central node's representation (see algorithm 12 in A.3.2). Essentially, this module queries parts of the graph where it expects specific nodes to be and then uses the distance of the neighbouring nodes to its query point to collect information from the other nodes.

Since the three main modules are applied sequentially in eight layers, the representations from each module allow the other modules to gradually extract more information from their inputs. For example, using the cryo-EM density, the network is able to find a better orientation for its backbone as well as a more accurate set of probabilities for its amino acid identity, which lets it search the sequence more accurately with the sequence attention module. This process of improvement continues while the positions of the atoms also get optimized using these representations through the application of the Backbone Frame module.

The Backbone Frame module (seen in figure 2B) takes as input the representation of each graph node that is the result of the sequential operation of the three main modules described above and outputs three vectors that describe the change in position of the $C^\alpha$, C, and N atom positions with respect to the network's current backbone affine frame. The shift in position is applied to the backbone atoms and the new backbone affine frame is calculated using Gram-Schmidt, similar to Algorithm 21 in AlphaFold2 (Jumper et al., 2021). Because the three shifts may distort the geometry of the peptide plane, the new coordinates for the backbone are calculated by aligning a peptide plane with ideal geometry (from literature bonds) with the shifted positions (see algorithm 16 in A.3.2).

### 3.2.2 TRAINING

Multiple loss functions define the tasks of the different modules and the resulting losses are optimized jointly with gradient descent. Most losses are calculated at each intermediate layer of the GNN so that it is able to learn the correct structure as early in the layers as possible. The most important losses are $C^\alpha$ root mean squared deviation (RMSD) loss; backbone RMSD loss; amino-acid classification loss; local confidence score loss; torsion angles loss; and full atom loss. A full definition of all losses is given in Appendix A.1.

The main training loop consists of taking a PDB structure, extracting just the $C^\alpha$ atoms, distorting them with noise, initializing the backbone frames for each node randomly, and then having the network predict the original PDB structure. Cryo-EM maps are augmented with the addition of noise and sharpening/dampening, similar to the training of the graph initialization network as described in section 3.1.3. Because the initial $C^\alpha$ positions are noisy, with an RMSD to the deposited model of 0.9 Å on average, one important loss function is

$$\mathcal{L}_{C^\alpha} = \frac{1}{N} \sum_i \text{RMSD}(\mathbf{x}_i, g_\phi(\mathbf{x}_i + \mathbf{e}_i)) \tag{3}$$

where $\mathbf{x}_i \in \mathbb{R}^3$ are the true $C^\alpha$ positions from the dataset, $g_\phi$ is the graph neural network, and $\mathbf{e}_i \sim \mathcal{N}(0, \frac{1}{\sqrt{3}})$. Note that $\mathbb{E}\|\mathbf{e}_i\|_2 \approx 0.9$ (this comes from the average norm of a Gaussian distributed vector). Denoising node positions has been shown to be a powerful training paradigm in other use cases as well (e.g. Godwin et al., 2021). In addition to noise added to the $C^\alpha$ positions, sometimes the graph initialization step misses some residues or adds extra ones. In order to have the GNN be able to deal with these scenarios, during training 10% of the residues are randomly removed and replaced with randomly generated peptides that are between 2-5 residues long. The network is then also trained to be able to predict whether or not a node actually exists in the model, or if it is extra. All classification losses use focal loss (Lin et al., 2017b). This includes the amino acid classification loss, the sequence match loss, and the edge prediction loss.

An important feature of this network is that it also gives a measure of its confidence in its output per residue. This is trained by having the network predict its backbone loss per residue. This output (referred to in 3.2 as $O$) is then normalized and saved in the B-factor section of the mmCIF file it outputs. This output is useful in pruning regions of the model where the network is not confident in the postprocessing step (see section 3.3). Generally, we observe that well-ordered, high-resolution parts of the cryo-EM map have higher confidence values than regions with disordered and lower resolution (see figure 2C).

The side chain atoms are generated through the prediction of their rotatable torsion angles with respect to the backbone frame (for an example, see figure 1C). We noticed better results if the network predicted torsion angles for all 20 possible amino acid assignments for each residue. Then, we index into the torsion angle predictions for each residue and pick the set of angles that correspond to the predicted amino acid. To train this part of the model, for each layer, the mean squared loss of the torsion angles of the target amino acid against the true torsion angles is calculated, and at the last layer, the all-atom RMSD to the target structure is also calculated.

### 3.2.3 RECYCLING

The output of one round of the GNN denoises the positions of the $C^\alpha$ nodes and gives better orientations for the backbone frames, and we observed that a subsequent round of the GNN, starting from the output of the previous round, improved the results further. We, therefore, train the GNN with recycling. For every training step, we randomly pick an integer $r \in \{1, 2, 3\}$ and run the GNN $r - 1$ times with gradients turned off, and then use the output to run the GNN one more time with gradients. This allows the GNN to learn to keep the input approximately unchanged when the positions are correct. We do not recycle the GNN features so they are recalculated with the corrected positions and orientations. The same recycling scheme, but with $r = 3$, is also used during inference.

## 3.3 POSTPROCESSING

The GNN processes the $C^\alpha$ atoms into a set of unordered residues. Next, we connect the residues into chains that define the full atomic model. In the strictest sense, not even the direction of the chains is defined by the GNN. However, using the fact that $\|C_{t-1} - N_t\|_2 < 1.4$ Å (known as peptide bonds, see figure 1A), we can combine the atomic coordinates predicted by the network as well as the edge prediction probabilities as a heuristic to connect residues. More concretely, the residues are tied so that the sum of peptide bond lengths across all nodes is minimized, ignoring links where the edge prediction is below a threshold of 0.5.

After the chains are connected, we use the amino acid prediction probabilities to construct an HMM profile. We then perform a sequence search using HMMER (Mistry et al., 2013) against the given set of sequences of the model. The uncertainty of the predicted amino-acid probabilities $P$ can vary due to several factors, e.g. map resolution or characteristics of the sequence itself, which can limit an accurate mapping of the sequence onto the structure by the GNN. Due to this uncertainty, doing a probabilistic search against the sequence after postprocessing gives superior performance over just assigning the highest probability amino acid. After alignment with the sequence search, residues that correspond to a "match" state (as defined in Krogh et al., 1994) are mutated to the amino acids that exist in the sequence. Based on the sequence search, we also connect separate chains that should be connected depending on both the matched sequence gap and the proximity of the chains.

Lastly, chains shorter than 4 residues are pruned and the resulting coordinates are used as the input to the GNN network again. This process continues for 3 recycling iterations and the end result undergoes a final "relaxation" step that uses physical restraint-based losses to optimize the positions of the model atoms using an L-BFGS optimizer (Liu & Nocedal, 1989). This step mainly alleviates unnatural side chain distance violations and does not noticeably affect the distance metrics in section 4, which are all based on main chain atoms.

## 4 RESULTS

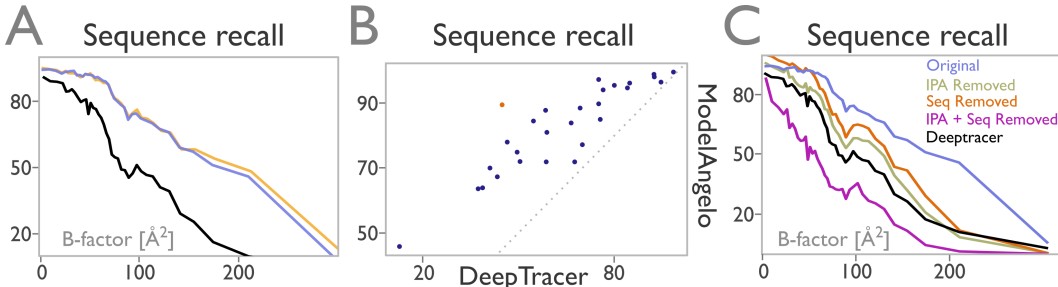

Figure 3: **(A)** shows sequence recall for all residues in the test dataset as a function of B-factor labels for final model outputs (after postprocessing), for Deeptracer (black) and ModelAngelo (before pruning in orange; after pruning in purple). **(B)** shows the same results but averaged for each PDB entry, with ModelAngelo's pruned prediction (y-axis) versus Deeptracer (x-axis). The dotted line marks the identity line. The orange marker represents PDB entry 8DTM, which is shown in figure 4. **(C)** shows the result of the ablation experiment, similar in format to (A).

Here we report results for a test dataset of 28 map-model pairs that were deposited to the PDB and EMDB after the cutoff date for training. Results on a smaller test set without protein chains with more than 30 % sequence similarity with any model in the training set are described in A.5. We implemented our approach in an open-source software package called ModelAngelo. Generally, the atomic models built by ModelAngelo are close to the deposited PDB structures and they degrade with the resolution of the cryo-EM map. The overall resolution of the 28 test maps ranges from 2.1 to 3.8 Å. However, flexibility in parts of the protein structures also leads to local variations in resolution across the maps. The latter is reflected in the refined B-factors of the individual residues in the deposited PDB coordinate files, where higher B-factors indicate lower local resolution. We compare our results against the current state-of-the-art method for automated model building, Deeptracer (Pfab et al., 2021). A comparison with results obtained with the Phenix software (Terwilliger et al., 2018), which performs worse than DeepTracer, is available in table 4 of A.2.

Our main metric is sequence recall: the percentage of residues for which the $C^\alpha$ atom is within 3 Å of the deposited model, and the amino acid prediction is correct. Figure 3 compares the performance of ModelAngelo and Deeptracer for each of the structures in the test dataset, and as a function of B-factor averaged over all residues. A more comprehensive list of metrics, for each of the 28 map-model pairs, is shown in Appendix A.2. Over the entire test dataset, there is little difference in sequence recall between the unpruned and the pruned model from ModelAngelo, which implies that pruning removes incorrectly built parts of the model.

Ablation studies (figure 3C) show that the improved results of ModelAngelo versus Deeptracer are because ModelAngelo is able to combine different modalities of information to build the model, rather than just the cryo-EM map. Removing the sequence module or the IPA module results in worse results that are still better than Deeptracer. However, removing both of these modules, such that the GNN only relies on the Cryo-EM module, results in predictions that are worse than Deeptracer.

Figure 4 illustrates the quality of the built models from ModelAngelo and Deeptracer for one example from the test dataset (PDB entry 8DTM); more examples are given in A.6. Because of its increased complexity, ModelAngelo is considerably slower than Deeptracer. Still, execution times for the test dataset are in the range of several minutes to one hour and a half, depending on the size of the structure (see A.4.3). Given that manual *de novo* model building takes on the order of weeks, we do not believe this to be a serious drawback.

## 5 DISCUSSION

Our results illustrate that combining the voxel-based information from the cryo-EM map with sequence information and graph topology is useful for automating the intensive task of atomic modelling in cryo-EM maps. Below, we consider the limitations of the current implementation of ModelAngelo, and outline lines of future research to overcome them.

**Sensitivity to resolution**. Even though the network has multiple different modalities of input, relatively low resolutions of the cryo-EM map will affect the results. The graph initialization by the CNN, and the amino acid classification that provides the information for mapping the sequence onto the main chain, are obvious examples that benefit from higher-resolution maps. Poor amino acid classifications may also lead to errors in the sequence assignment in the postprocessing step, which may then feed into the subsequent HMM sequence alignment and lead to incorrect chain assignments. This is more likely to happen for complexes with many similar sequences. In practice, we observe that ModelAngelo's performance starts degrading at resolutions worse than 3.5 Å, see Appendix A.3. A similar trend is also typical for manual model building. It may be possible to combine the embedding of information from relatively low-resolution cryo-EM maps (e.g 10 Å) with methods for protein structure prediction, such as AlphaFold2 (Jumper et al., 2021), which would extend beyond what is possible for manual building. Despite the observation that ModelAngelo already uses some ideas from AlphaFold2, such an approach for building in low-resolution maps, which would blur the boundaries between experimental structure determination and prediction, would require major changes to the approaches outlined in this paper.

**Nucleic acids**. Many large complexes that are solved with cryo-EM comprise nucleic acids as well as proteins, e.g. ribonucleic acids (RNA) in spliceosomes and ribosomes, or deoxyribonucleic acids

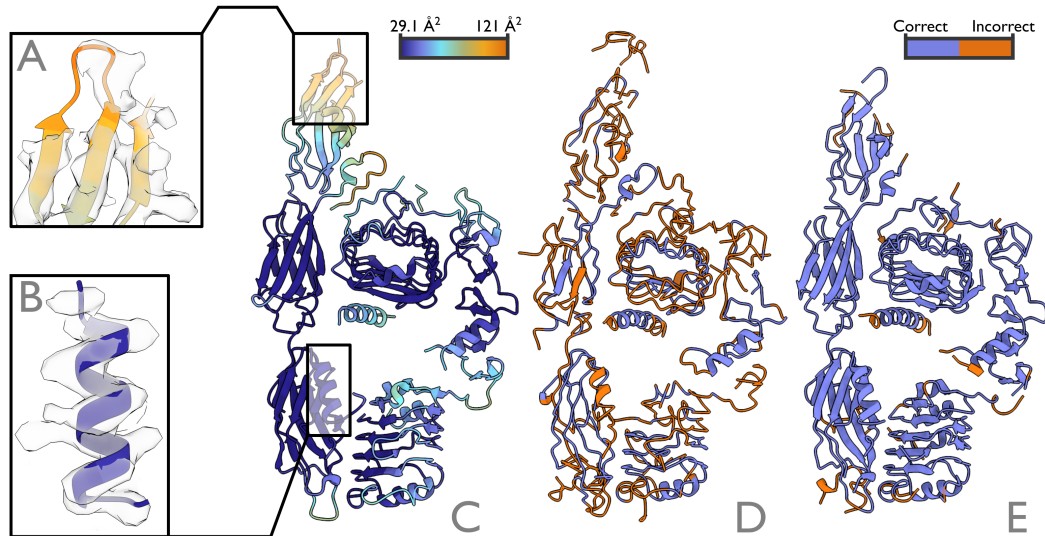

Figure 4: Comparison of the deposited model for PDB entry 8DTM **(C)**, Deeptracer's prediction **(D)** and ModelAngelo's pruned model **(E)**. The deposited model is coloured according to the refined B-factor. Meanwhile, the predictions are coloured orange where their amino acid prediction is different from the deposited structure, and purple where it is the same. **(A)** shows an iso-surface of the cryo-EM map and the deposited model for a high B-factor region, with correspondingly poor density. **(B)** shows the same for a low B-factor region, where side chains are well resolved in the density.

(DNA) in replication or transcription machinery. The backbone of DNA or RNA strands is made from alternating phosphate and sugar groups. The phosphorus atoms have high contrast in the cryo-EM map, which makes the segmentation problem of nucleic acids easier than that of protein residues. However, the main difficulty lies in identifying the correct sequence for the nucleobases that make up the equivalent of side chains for RNA or DNA strands. There are four typical bases for both RNA and DNA: two purines and two pyrimidines. At resolutions around 3.5 Å, one can distinguish the purines from the pyrimidines, but not the two purines or pyrimidines from each other. This makes sequencing DNA or RNA strands in 3.5 Å cryo-EM maps difficult. Yet, already the ability to automatically model nucleotide backbones, together with a classifier to distinguish purines from pyrimidines, would alleviate the task of manual model building. Therefore, we plan to add support for building RNA or DNA to ModelAngelo in the near future.

**Unknown sequences**. Because cryo-EM can be performed on samples that are extracted from native cells or tissues, it is not always obvious which proteins are present in a cryo-EM map, (for an example, see Schweighauser et al., 2022). Our current implementation depends on a user-provided sequence file that defines all proteins present in the map. Recently, semi-automated tools to identify proteins in cryo-EM maps have been reported. For example, findMySequence (Chojnowski et al., 2022) can search protein sequence databases, using amino acid classifications after a backbone model has been built in the cryo-EM map. We plan to extend our approach with a sequence-free model to fully automate this process, by combining automated model building with searches through large sequence databases like Uniclust (Mirdita et al., 2017) using our HMM search procedure combined with tools such as HHblits (Remmert et al., 2012).

# 6 ACKNOWLEDGEMENTS

We thank Johannes Schwab, Joe Greener, Sofia Lövestam, David Li, Keitaro Yamashita, Garib Murshudov, Carola-Bibiane Schönlieb, Tanmay Bharat, Zo Ford, and Rafael Fernández Leiro for helpful discussions, and Jake Grimmett, Toby Darling, and Ivan Clayson for help with high-performance computing. This work was supported by the Medical Research Council (MC_UP_A025_1013 to S.H.W.S.).

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

# A  Appendix

## A.1  Losses

This is a full description of all the losses used for training the GNN. These losses can be divided into two groups: one-step losses and auxiliary losses. One-step losses are losses that are only defined once for the final step of the GNN. However, auxiliary losses are summed over each step of the GNN, though the last step has a higher weighting.

### A.1.1  Auxiliary Losses

These losses are as follows: $C^\alpha$ RMSD loss, backbone frame loss, amino-acid classification loss, edge classification loss, local confidence score loss, and existence loss.

The $C^\alpha$ **RMSD loss** is just the RMSD of the $C^\alpha$ atoms only. That is

$$\mathcal{L}_{C^\alpha} = \frac{1}{N} \sum_i \mathrm{RMSD}(\mathbf{x}_i, g_\theta(\mathbf{x}_i + \mathbf{e}_i))$$

$$= \frac{1}{N} \sum_i \sqrt{\frac{1}{3} \sum_{d=1}^{3} \left| [\mathbf{x}_i]_d - [g_\theta(\mathbf{x}_i + \mathbf{e}_i)]_d \right|^2}$$

The **backbone frame loss** is the RMSD of the three backbone atoms that define the backbone frame: $C^\alpha$, C, and N atoms. The over-representation of the RMSD of the $C^\alpha$ atoms, through its contribution to two losses, is intended.

The **amino-acid classification loss** is the focal loss between the amino-acid classification logits and the target amino-acid recorded for each node. We have observed that even after the convergence of the GNN, the auxiliary loss of this classification task is relatively high, meaning that earlier layers are not well capable of distinguishing amino acids. This could be because the $C^\alpha$ positions are still noisy or that the network needs to pool neighbouring information more before it can be confident about the classification.

The **edge classification loss** is based on whether two ground truth $C^\alpha$ nodes are within 3.9 Å of each other. If they are, then we expect both nodes $a$ and $b$ to classify each other as a neighbour. Otherwise, both need to classify the other as not a neighbour.

The **local confidence score loss** is a simple regression loss that tries to predict the backbone frame loss described above for each residue. This gives a good measure of how confident the network is in the position of the residue. This measure is then normalized between 0 and 100 based on a simple heuristic so that larger expected losses lead to lower confidence values.

Finally, the **existence loss** is a focal loss that classifies whether a residue is an artificially added residue or it actually exists.

### A.1.2 ONE-STEP LOSSES

The **full-atom loss** is the same RMSD loss defined above for the $C^\alpha$ atom, except that it is calculated for all atoms, including side chain atoms.

The **torsion angles loss** is defined as the $L_2$ distance between the torsion angles of the network output and the torsion angles of the deposited model. A small loss is added to make sure the norm of the torsion angles is equal to 1. This is similar to the loss defined in section 1.9.1 of Jumper et al. (2021).

The **sequence attention loss** is a classification loss defined over the length of the given sequence for each residue, designed to match the position of the graph node. It is calculated for the last attention head of the sequence attention module as a focal loss over the calculated attention score.

The **violation loss** is meant to constrain the network to output models that are more in line with ideal bond lengths and angles. This loss is the same as defined in section 1.9.11 in Jumper et al. (2021).

### A.2 DETAILED METRICS PER PDB ENTRY IN THE TEST DATASET

The tables below contain the resolution (in Å) and various metrics for each of the 28 PDB entries in the test dataset. Backbone RMSD and sequence recall are defined in the main text. $C^\alpha$ recall is the percentage of residues for which the $C^\alpha$ atom is within 3 Å of the deposited model; $C^\alpha$ precision is the percentage of $C^\alpha$ atoms predicted that are within 3 Å of a $C^\alpha$ atom in the deposited map; lDDT is the $C^\alpha$ local distance difference test as described in Mariani et al. (2013); and sequence match is the percentage of matching amino-acids for the corresponding residues. Table 1 describes the results obtained by Deeptracer; tables 2 and 3 describe the results obtained by ModelAngelo, after and before pruning, respectively.

| PDB | Reso- lution | backbone RMSD | $C^\alpha$ recall | $C^\alpha$ precision | lDDT | Sequence match | Sequence recall |
|---|---|---|---|---|---|---|---|
| 7tu5 | 2.1 | 0.591 | 94.3 | 99.4 | 96.7 | 90.0 | 84.8 |
| 7unl | 2.45 | 0.496 | 97.3 | 98.2 | 97.4 | 95.0 | 92.4 |
| 7q1u | 2.7 | 0.546 | 99.3 | 99.2 | 96.2 | 99.2 | 98.5 |
| 7szi | 2.7 | 0.635 | 98.3 | 99.0 | 94.2 | 96.3 | 94.6 |
| 7txv | 2.7 | 0.837 | 76.3 | 97.8 | 91.2 | 77.3 | 58.9 |
| 7uck | 2.8 | 0.866 | 87.8 | 49.0 | 93.6 | 85.5 | 75.1 |
| 7ode | 2.84 | 0.960 | 66.6 | 29.0 | 93.1 | 69.9 | 46.5 |
| 7sba | 2.9 | 0.829 | 88.4 | 93.9 | 92.2 | 75.2 | 66.5 |
| 7ugg | 3.16 | 0.896 | 93.0 | 98.7 | 93.2 | 82.3 | 76.5 |
| 7pt6 | 3.2 | 0.750 | 92.4 | 99.0 | 93.3 | 91.1 | 84.1 |
| 7xpx | 3.2 | 0.762 | 82.0 | 58.8 | 96.1 | 92.2 | 75.6 |
| 7px8 | 3.27 | 0.637 | 97.6 | 99.2 | 94.0 | 94.7 | 92.5 |
| 7sr8 | 3.3 | 1.026 | 95.4 | 95.3 | 86.6 | 78.8 | 75.1 |
| 7wug | 3.3 | 0.778 | 97.1 | 95.8 | 92.5 | 82.5 | 80.0 |
| 7y9u | 3.3 | 0.739 | 82.9 | 98.7 | 94.3 | 84.5 | 70.0 |
| 7u50 | 3.4 | 0.734 | 76.6 | 88.3 | 92.9 | 88.3 | 67.6 |
| 7sjn | 3.4 | 1.213 | 70.5 | 95.5 | 85.7 | 53.1 | 37.4 |
| 7z1m | 3.4 | 0.954 | 92.9 | 97.7 | 89.2 | 74.6 | 69.3 |
| 7rzy | 3.5 | 1.544 | 71.4 | 94.2 | 79.7 | 17.9 | 12.8 |
| 7w9l | 3.5 | 0.977 | 80.6 | 96.1 | 90.2 | 72.8 | 58.7 |
| 8dtm | 3.5 | 1.420 | 95.9 | 81.6 | 87.1 | 46.8 | 44.9 |
| 8a3t | 3.5 | 1.380 | 76.5 | 96.6 | 81.1 | 50.8 | 38.8 |
| 8a2q | 3.53 | 1.260 | 74.0 | 96.6 | 85.2 | 55.5 | 41.1 |
| 7yzk | 3.57 | 1.293 | 94.8 | 88.4 | 82.7 | 53.2 | 50.5 |
| 7oix | 3.6 | 1.339 | 95.1 | 77.9 | 82.0 | 45.7 | 43.4 |
| 7tvz | 3.6 | 1.261 | 91.8 | 96.3 | 85.9 | 59.6 | 54.7 |
| 7pt7 | 3.8 | 1.202 | 86.3 | 96.4 | 86.5 | 57.5 | 49.6 |
| 7um0 | 3.8 | 1.102 | 86.2 | 98.1 | 86.7 | 67.8 | 58.5 |

Table 1: Deeptracer Results

| PDB | Reso-lution | backbone RMSD | $C^\alpha$ recall | $C^\alpha$ precision | lDDT | Sequence match | Sequence recall |
|---|---|---|---|---|---|---|---|
| 7tu5 | 2.1 | 0.287 | 96.3 | 99.7 | 98.7 | 99.9 | 96.2 |
| 7unl | 2.45 | 0.310 | 99.2 | 94.7 | 98.9 | 99.7 | 98.9 |
| 7q1u | 2.7 | 0.268 | 99.6 | 99.3 | 99.2 | 99.9 | 99.6 |
| 7szi | 2.7 | 0.398 | 97.2 | 99.4 | 98.4 | 99.3 | 96.5 |
| 7txv | 2.7 | 0.686 | 89.5 | 91.5 | 92.3 | 90.5 | 81.0 |
| 7uck | 2.8 | 0.385 | 97.6 | 99.5 | 97.4 | 99.8 | 97.3 |
| 7ode | 2.84 | 0.418 | 81.5 | 99.2 | 96.7 | 95.7 | 78.0 |
| 7sba | 2.9 | 0.444 | 86.1 | 98.2 | 96.1 | 97.4 | 83.9 |
| 7ugg | 3.16 | 0.468 | 96.3 | 98.6 | 95.9 | 97.7 | 94.1 |
| 7pt6 | 3.2 | 0.435 | 95.7 | 99.1 | 96.6 | 98.9 | 94.7 |
| 7xpx | 3.2 | 0.343 | 85.4 | 97.1 | 97.6 | 99.5 | 85.0 |
| 7px8 | 3.27 | 0.368 | 99.0 | 98.2 | 97.7 | 99.1 | 98.1 |
| 7sr8 | 3.3 | 0.537 | 92.7 | 98.0 | 93.5 | 96.9 | 89.8 |
| 7wug | 3.3 | 0.396 | 96.2 | 94.1 | 97.0 | 99.3 | 95.5 |
| 7y9u | 3.3 | 0.388 | 77.9 | 99.0 | 97.7 | 99.1 | 77.2 |
| 7u50 | 3.4 | 0.396 | 73.7 | 97.7 | 97.0 | 97.6 | 71.9 |
| 7sjn | 3.4 | 0.740 | 68.3 | 96.3 | 90.5 | 93.1 | 63.6 |
| 7z1m | 3.4 | 0.472 | 90.4 | 99.3 | 95.3 | 97.9 | 88.5 |
| 7rzy | 3.5 | 0.746 | 47.2 | 99.2 | 90.1 | 97.2 | 45.8 |
| 7w9l | 3.5 | 0.447 | 74.0 | 98.7 | 96.2 | 97.2 | 71.9 |
| 8dtm | 3.5 | 0.557 | 92.0 | 90.9 | 93.8 | 97.3 | 89.5 |
| 8a3t | 3.5 | 0.706 | 66.9 | 98.6 | 90.4 | 95.6 | 63.9 |
| 8a2q | 3.53 | 0.644 | 74.1 | 94.8 | 90.8 | 94.4 | 70.0 |
| 7yzk | 3.57 | 0.811 | 81.0 | 94.4 | 88.5 | 88.9 | 72.0 |
| 7oix | 3.6 | 0.636 | 75.1 | 96.4 | 91.2 | 89.6 | 67.3 |
| 7tvz | 3.6 | 0.563 | 86.5 | 98.3 | 93.3 | 97.7 | 84.5 |
| 7pt7 | 3.8 | 0.476 | 78.2 | 99.5 | 95.0 | 95.9 | 74.9 |
| 7um0 | 3.8 | 0.618 | 89.8 | 99.0 | 92.2 | 97.7 | 87.8 |

Table 2: ModelAngelo Pruned Results

| PDB | Reso-lution | backbone RMSD | $C^\alpha$ recall | $C^\alpha$ precision | lDDT | Sequence match | Sequence recall |
|---|---|---|---|---|---|---|---|
| 7tu5 | 2.1 | 0.307 | 99.7 | 98.1 | 98.3 | 96.7 | 96.5 |
| 7unl | 2.45 | 0.311 | 99.9 | 84.8 | 98.8 | 98.9 | 98.8 |
| 7q1u | 2.7 | 0.273 | 100.0 | 97.7 | 99.1 | 99.6 | 99.6 |
| 7szi | 2.7 | 0.406 | 99.1 | 98.2 | 98.3 | 97.8 | 96.9 |
| 7txv | 2.7 | 0.710 | 92.3 | 87.0 | 91.9 | 88.4 | 81.6 |
| 7uck | 2.8 | 0.401 | 99.0 | 98.2 | 97.2 | 98.4 | 97.4 |
| 7ode | 2.84 | 0.549 | 96.9 | 93.5 | 94.6 | 84.2 | 81.7 |
| 7sba | 2.9 | 0.516 | 97.4 | 91.5 | 94.7 | 87.6 | 85.3 |
| 7ugg | 3.16 | 0.495 | 98.9 | 97.1 | 95.4 | 95.4 | 94.3 |
| 7pt6 | 3.2 | 0.466 | 98.1 | 97.7 | 96.1 | 96.6 | 94.9 |
| 7xpx | 3.2 | 0.563 | 98.2 | 90.3 | 93.7 | 88.1 | 86.5 |
| 7px8 | 3.27 | 0.371 | 99.9 | 96.5 | 97.7 | 98.5 | 98.4 |
| 7sr8 | 3.3 | 0.574 | 99.0 | 93.7 | 92.7 | 92.0 | 91.1 |
| 7wug | 3.3 | 0.420 | 99.8 | 82.0 | 96.5 | 96.1 | 95.9 |
| 7y9u | 3.3 | 0.719 | 99.1 | 86.0 | 91.7 | 82.8 | 82.1 |
| 7u50 | 3.4 | 0.848 | 99.1 | 67.2 | 89.6 | 76.3 | 75.6 |
| 7sjn | 3.4 | 1.071 | 98.6 | 78.7 | 85.8 | 70.2 | 69.2 |
| 7z1m | 3.4 | 0.546 | 98.0 | 97.6 | 93.9 | 91.4 | 89.6 |
| 7rzy | 3.5 | 1.122 | 95.3 | 85.9 | 82.8 | 57.2 | 54.5 |
| 7w9l | 3.5 | 0.646 | 90.2 | 89.2 | 92.6 | 83.0 | 74.9 |
| 8dtm | 3.5 | 0.649 | 99.5 | 74.8 | 92.1 | 90.5 | 90.0 |
| 8a3t | 3.5 | 0.960 | 86.5 | 94.3 | 85.9 | 77.7 | 67.3 |
| 8a2q | 3.53 | 0.923 | 98.3 | 82.7 | 86.6 | 74.7 | 73.5 |
| 7yzk | 3.57 | 0.935 | 99.4 | 75.4 | 86.2 | 74.4 | 73.9 |
| 7oix | 3.6 | 0.843 | 99.4 | 74.9 | 87.6 | 71.1 | 70.7 |
| 7tvz | 3.6 | 0.661 | 98.5 | 93.2 | 91.5 | 87.8 | 86.5 |
| 7pt7 | 3.8 | 0.583 | 88.9 | 97.4 | 93.2 | 86.2 | 76.6 |
| 7um0 | 3.8 | 0.727 | 98.6 | 93.8 | 90.5 | 90.5 | 89.2 |

Table 3: ModelAngelo Unpruned Results

| PDB | backbone RMSD | $C^\alpha$ recall | $C^\alpha$ precision | lDDT | Sequence match | Sequence recall |
|------|------|------|------|------|------|------|
| 7tu5 | 0.841 | 60.5 | 95.6 | 85.1 | 58.8 | 35.5 |
| 7unl | 0.692 | 88.1 | 96.4 | 89.5 | 80.7 | 71.1 |
| 7q1u | 0.862 | 89.5 | 94.5 | 84.7 | 70.5 | 63.1 |
| 7szi | 0.843 | 85.7 | 96.4 | 88.1 | 81.7 | 70.0 |
| 7txv | 1.028 | 44.6 | 94.3 | 83.5 | 62.8 | 28.0 |
| 7ode | 1.103 | 8.0 | 51.9 | 84.9 | 25.7 | 2.0 |
| 7sba | 1.127 | 64.7 | 93.6 | 80.8 | 28.4 | 18.4 |
| 7ugg | 1.277 | 59.5 | 94.6 | 79.1 | 36.7 | 21.9 |
| 7xpx | 0.889 | 69.6 | 86.5 | 83.5 | 40.8 | 28.4 |
| 7px8 | 1.055 | 83.0 | 90.4 | 80.5 | 60.5 | 50.3 |
| 7sr8 | 1.833 | 45.4 | 94.0 | 74.6 | 7.7 | 3.5 |
| 7wug | 1.027 | 82.7 | 95.4 | 83.5 | 57.7 | 47.7 |
| 7y9u | 1.068 | 67.5 | 95.7 | 81.8 | 40.8 | 27.5 |
| 7u50 | 1.049 | 56.5 | 79.1 | 81.0 | 36.6 | 20.7 |
| 7sjn | 1.580 | 33.0 | 95.8 | 78.3 | 19.6 | 6.5 |
| 7rzy | 2.103 | 24.0 | 92.1 | 70.1 | 8.8 | 2.1 |
| 7w9l | 1.177 | 53.3 | 95.7 | 80.5 | 47.6 | 25.4 |
| 8dtm | 1.808 | 48.8 | 87.5 | 74.8 | 11.3 | 5.5 |
| 8a2q | 1.503 | 53.0 | 95.3 | 75.1 | 34.5 | 18.3 |
| 7yzk | 1.575 | 56.8 | 92.1 | 77.3 | 27.0 | 15.4 |
| 7oix | 1.933 | 60.2 | 95.4 | 71.6 | 7.2 | 4.3 |
| 7tvz | 1.474 | 39.1 | 95.2 | 76.2 | 31.4 | 12.3 |
| 7um0 | 1.611 | 54.1 | 95.2 | 75.7 | 16.5 | 9.0 |

Table 4: PHENIX Results

## A.3 NETWORK DETAILS

### A.3.1 GRAPH INITIALIZATION NETWORK

---

**Algorithm 1** Segmentation Forward Pass. The fully convolutional segmentation model described in section 3.1.1.

---
1: **function** SEGMENTATION_FORWARD($V$)
2:     $V = (V - mean(V))/std(V)$
3:     ds_0, ds_1, ds_2, ds_3, ds_4 ← downsample_forward($V$)
4:     tl4 ← conv_building_block(ds_4, input_channels=512, output_channels=256)
5:     ll4 ← conv_building_block(ds_3, input_channels=256, output_channels=256)
6:     c4 ← main_layer(upsample_add(tl4, ll4), input_channels=256, expansion=4, num_layers=2)
7:     tl3 ← conv_building_block(c4, input_channels=1024, output_channels=128)
8:     ll3 ← conv_building_block(ds_2, input_channels=128, output_channels=128)
9:     c3 ← main_layer(upsample_add(tl3, ll3), input_channels=128, expansion=4, num_layers=20)
10:     tl2 ← conv_building_block(c3, input_channels=512, output_channels=64)
11:     ll2 ← conv_building_block(ds_1, input_channels=64, output_channels=64)
12:     c2 ← main_layer(upsample_add(tl2, ll2), input_channels=64, expansion=4, num_layers=50)
13:     tl1 ← conv_building_block(c2, input_channels=256, output_channels=64)
14:     ll1 ← conv_building_block(ds_0, input_channels=64, output_channels=64)
15:     c1 ← main_layer(upsample_add(tl1, ll1), input_channels=64, expansion=4, num_layers=10)
16:     pred ← multi_scale_conv(c1, input_channels=256, mid_channels=64, output_channels=1)
17:     return pred

---

**Algorithm 2** Convolutional Building Block

---
1: **function** CONV_BUILDING_BLOCK($f$, input_channels, output_channels)
2:     $f$ ← conv3d($f$, input_channels, output_channels, kernel_size=3, stride=1, padding=1)
3:     $f$ ← instance_norm_3d($f$, output_channels, affine=True)
4:     $f$ ← relu($f$)
5:     return $f$

---

**Algorithm 3** Bottleneck

---
1: **function** BOTTLENECK($i$, input_channels, hidden_channels, expansion)
2:     $f$ ← conv_building_block($i$, input_channels, hidden_channels)
3:     $f$ ← conv_building_block($f$, hidden_channels, hidden_channels)
4:     $f$ ← conv3d($f$, hidden_channels, hidden_channels $\times$ expansion, kernel_size=1, stride=1, padding=1)
5:     $f$ ← instance_norm_3d($f$, output_channels, affine=True)
6:     $f$ ← relu($f + i$)
7:     return $f$

---

---

**Algorithm 4** Downsample Layer. Factor-two downsampling with strided convolutions.

---
1: **function** DOWNSAMPLE(input_channels, output_channels, $\mathbf{f}$)
2:     $\mathbf{f} \leftarrow$ conv3d($\mathbf{f}$, input_channels, output_channels, kernel_size=3, stride=2, padding=1)
3:     $\mathbf{f} \leftarrow$ instance_norm_3d($\mathbf{f}$, output_channels, affine=True)
4:     $\mathbf{f} \leftarrow$ relu($\mathbf{f}$)
5:     return $\mathbf{f}$

---

**Algorithm 5** Downsample Forward Pass

---
1: **function** DOWNSAMPLE_FORWARD($\boldsymbol{V}$)
2:     ds_0 $\leftarrow$ conv_building_block($\boldsymbol{V}$, 1, 64, kernel_size=5, padding=2)
3:     ds_1 $\leftarrow$ downsample(ds_0, input_channels=64, output_channels=64)
4:     ds_2 $\leftarrow$ downsample(ds_1, input_channels=64, output_channels=128)
5:     ds_3 $\leftarrow$ downsample(ds_2, input_channels=128, output_channels=256)
6:     ds_4 $\leftarrow$ downsample(ds_3, input_channels=256, output_channels=512)
7:     return ds_0, ds_1, ds_2, ds_3, ds_4

---

**Algorithm 6** Main Layer

---
1: **function** MAIN_LAYER($\mathbf{f}$, input_channels, expansion, num_layers)
2:     hidden_channels $\leftarrow$ input_channels
3:     **for** $i$ in range(num_layers) **do**
4:         $\mathbf{f} \leftarrow$ bottleneck($\mathbf{f}$, input_channels, hidden_channels, expansion)
5:         input_channels $\leftarrow$ hidden_channels $\times$ expansion
6:     return $\mathbf{f}$

---

**Algorithm 7** Upsample then add

---
1: **function** UPSAMPLE_ADD($\mathbf{f}$, $\mathbf{g}$)
2:     $H, W, D \leftarrow$ return_shape($\mathbf{g}$)
3:     upsampled $\leftarrow$ trilinear_interpolation($\mathbf{f}$, $H, W, D$)
4:     return upsampled + $\mathbf{g}$

---

**Algorithm 8** Multi Scale Convolution

---
1: **function** MULTI_SCALE_CONV($\mathbf{f}$, input_channels, mid_channels, output_channels)
2:     $\mathbf{f}_3 \leftarrow$ conv3d($\mathbf{f}$, input_channels, mid_channels, kernel_size=3, stride=1, padding=1)
3:     $\mathbf{f}_5 \leftarrow$ conv3d($\mathbf{f}$, input_channels, mid_channels, kernel_size=5, stride=1, padding=2)
4:     $\mathbf{f}_7 \leftarrow$ conv3d($\mathbf{f}$, input_channels, mid_channels, kernel_size=7, stride=1, padding=3)
5:     $\mathbf{f} \leftarrow$ concatenate($\mathbf{f}_3, \mathbf{f}_5, \mathbf{f}_7$)
6:     $\mathbf{f} \leftarrow$ instance_norm_3d($\mathbf{f}$, mid_channels $\times$ 3, affine=True)
7:     $\mathbf{f} \leftarrow$ relu($\mathbf{f}$)
8:     $\mathbf{f} \leftarrow$ conv3d($\mathbf{f}$, mid_channels $\times$ 3, output_channels, padding=1, kernel_size=3)
9:     return $\mathbf{f}$

---

A.3.2   GRAPH NEURAL NETWORK

---

**Algorithm 9** GNN Forward Pass

---
1: **function** GNN_FORWARD_PASS($\mathbf{x}, \mathbf{F}, \boldsymbol{V}, \boldsymbol{S}$, num_recycling_steps)
2:     **for** $k$ in range(num_recycling_steps) **do**
3:         $\mathbf{f} \leftarrow$ zeros(batch_size, 256)
4:         **for** $i$ in range(num_layers=8) **do**
5:             $\mathbf{f} \leftarrow$ cryo_em_attention($\mathbf{x}, \mathbf{f}, \mathbf{F}, \boldsymbol{V}$)
6:             $\mathbf{f} \leftarrow$ sequence_attention($\mathbf{f}, \mathbf{S}$)
7:             $\mathbf{f} \leftarrow$ spatial_ipa($\mathbf{x}, \mathbf{f}, \mathbf{F}$)
8:             $\mathbf{f} \leftarrow$ transition_layer($\mathbf{f}$)
9:             $\mathbf{P} \leftarrow$ amino_acid_classification($\mathbf{f}$)
10:            $\mathbf{O} \leftarrow$ local_confidence_prediction($\mathbf{f}$)
11:            $\mathbf{x}, \mathbf{F} \leftarrow$ backbone_frame_module($\mathbf{f}, \mathbf{F}$)
12:        $\mathbf{G} \leftarrow$ torsion_angle_network($\mathbf{f}$)
13:    **return** $\mathbf{x}, \mathbf{F}, \mathbf{P}, \mathbf{O}, \mathbf{G}$

---

**Algorithm 10** CryoEM Attention Module

---
1: **function** CRYO_EM_ATTENTION($\mathbf{x}, \mathbf{z}, \mathbf{F}, \boldsymbol{V}$)
2:     $\boldsymbol{C} \leftarrow$ get_cubes_centered_on_nodes($\mathbf{x}, \mathbf{F}, \boldsymbol{V}$)
3:     $\boldsymbol{R} \leftarrow$ get_rectangles_between_nodes($\mathbf{x}, \mathbf{F}, \boldsymbol{V}$)
4:     $\mathbf{d} \leftarrow$ generate_distance_based_features($\mathbf{x}$)
5:     $\mathbf{n}_e \leftarrow [\mathbf{z}, \mathbf{d}]_e$
6:     $\mathbf{k} \leftarrow$ get_cryo_key_vectors($\boldsymbol{R}$)
7:     $\mathbf{q}, \mathbf{v} \leftarrow$ get_cryo_query_value_vectors($\mathbf{n}_e$)
8:     $\mathbf{z}_c \leftarrow$ softmax($\mathbf{q}^T \mathbf{k}$) $\cdot \mathbf{v}$
9:     $\mathbf{z}_p \leftarrow$ get_cryo_point_features($\boldsymbol{C}$)
10:    $\mathbf{z}' \leftarrow$ linear_layer($[\mathbf{z}_c, \mathbf{z}_p]$, output_dim=256)
11:    $\mathbf{z}' \leftarrow$ dropout($\mathbf{z}'$, probability=0.5)
12:    $\mathbf{z} \leftarrow$ layer_norm($\mathbf{z} + \mathbf{z}'/\sqrt{2}$)
13:    **return** $\mathbf{z}$

---

**Algorithm 11** Sequence Attention Module

---
1: **function** SEQUENCE_ATTENTION($\mathbf{f}, \mathbf{S}$)
2:     $\mathbf{Q} \leftarrow$ linear_layer($\mathbf{f}$, output_dim=$8 \times 256$)
3:     $\mathbf{Q} \leftarrow$ reshape($\mathbf{Q}$, shape=(batch_size, attention_heads=8, feature_dim=256))
4:     $\mathbf{V} \leftarrow$ linear_layer($\mathbf{S}$, output_dim=$8 \times 256$)
5:     $\mathbf{V} \leftarrow$ reshape($\mathbf{V}$, shape=(sequence_len, attention_heads=8, feature_dim=256))
6:     $\mathbf{K} \leftarrow$ linear_layer($\mathbf{K}$, output_dim=$8 \times 256$)
7:     $\mathbf{K} \leftarrow$ reshape($\mathbf{K}$, shape=(sequence_len, attention_heads=8, feature_dim=256))
8:     $\mathbf{z} \leftarrow$ softmax($\mathbf{Q}^T \mathbf{K}$) $\cdot \mathbf{V}$
9:     $\mathbf{z} \leftarrow$ reshape($\mathbf{z}$, shape=(batch_size, $8 \times 256$))
10:    $\mathbf{z} \leftarrow$ layer_norm($\mathbf{z}$)
11:    $\mathbf{z} \leftarrow$ linear_layer($\mathbf{z}$, output_dim=256)
12:    $\mathbf{z} \leftarrow$ dropout($\mathbf{z}$, probability=0.5)
13:    $\mathbf{f} \leftarrow$ layer_norm($\mathbf{f} + \mathbf{z}/\sqrt{2}$)
14:    **return** $\mathbf{f}$

---

---

**Algorithm 12** Spatial IPA Module

---

1: **function** SPATIAL_IPA($\mathbf{x}$, $\mathbf{z}$, $\mathbf{F}$)
2:     $\mathbf{d} \leftarrow$ generate_distance_based_features($\mathbf{x}$)
3:     $\mathbf{e} \leftarrow$ get_nearest_twenty_neighbours($\mathbf{x}$)
4:     $\mathbf{n}_e \leftarrow [\mathbf{z}, \mathbf{d}]_e$
5:     $\mathbf{v} \leftarrow$ get_value_vector($\mathbf{n}_e$)
6:     $\mathbf{q} \leftarrow$ get_query_vector_in_local_frame($\mathbf{n}_e$)
7:     $\mathbf{q} \leftarrow \mathbf{F} \circ \mathbf{q}$                                                $\triangleright$ Bring to global frame
8:     $\mathbf{x}_n \leftarrow$ get_neighbour_positions($\mathbf{x}$)                  $\triangleright$ Shape: $M \times k \times 3$
9:     $\mathbf{z}' \leftarrow \text{softmax}(-\sum_{i \in [p]} \|\mathbf{q}_i - \mathbf{x}_n\|^2) \cdot \mathbf{v}$        $\triangleright$ Sum is over query points
10:    $\mathbf{z}' \leftarrow$ linear_layer($\mathbf{z}'$, output_dim=256)
11:    $\mathbf{z}' \leftarrow$ dropout($\mathbf{z}'$, probability=0.5)
12:    $\mathbf{z} \leftarrow$ layer_norm($\mathbf{z} + \mathbf{z}'/\sqrt{2}$)
13:    return $\mathbf{z}$

---

**Algorithm 13** Cubes Centered on Nodes

---

1: **function** GET_CUBES_CENTERED_ON_NODES($\mathbf{x}$, $\mathbf{F}$, $V$)
2:     affine_grid $\leftarrow$ torch.nn.functional.affine_grid($\mathbf{F}$, shape=(17, 17, 17))
3:     $C \leftarrow$ torch.nn.functional.grid_sample($V$, affine_grid)
4:     return $C$

---

**Algorithm 14** Rectangles Between Nodes

---

1: **function** GET_RECTANGLES_BETWEEN_NODES($\mathbf{x}$, $V$)
2:     $\mathbf{v} \leftarrow$ get_vectors_to_nearest_neighbours($\mathbf{x}$)
3:     $\mathbf{M} \leftarrow$ rotation_matrix_point_z_axis_to_vector($\mathbf{v}$)
4:     $\mathbf{F} \leftarrow$ concatenate($\mathbf{M}$, $\mathbf{x}$, dim=1)
5:     affine_grid $\leftarrow$ torch.nn.functional.affine_grid($\mathbf{F}$, shape=(5, 1, 1))
6:     $R \leftarrow$ torch.nn.functional.grid_sample($V$, affine_grid)
7:     return $R$

---

**Algorithm 15** Distance Based Features

---

1: **function** GENERATE_DISTANCE_BASED_FEATURES($\mathbf{f}$, $\mathbf{F}$)
2:     $\mathbf{x} \leftarrow \mathbf{F}[..., -1]$
3:     $\mathbf{e} \leftarrow$ get_k_nearest_neighbours($\mathbf{x}$, $k = 20$)                  $\triangleright$ Shape: $M \times k$
4:     neighbour_positions_in_local_frame $\leftarrow \mathbf{F}^{-1} \circ \mathbf{x}[\mathbf{e}]$      $\triangleright$ Shape: $M \times k \times 3$
5:     neighbour_distances $\leftarrow \|$neighbour_positions_in_local_frame$\|^2$    $\triangleright$ Shape: $M \times k$
6:     NCaC_pos $\leftarrow$ backbone_frame_to_pos($\mathbf{F}$)            $\triangleright$ See AlphaFold2 sup. 1.8.4
7:     CatoNCaC_distances $\leftarrow \|$NCaC_pos$[\mathbf{e}]$ - $\mathbf{x}\|^2$
8:     NtoC_distances $\leftarrow \|$NCaC_pos$[\mathbf{e}][..., 2]$ - NCaC_pos$[..., 0]\|^2$
9:     CtoN_distances $\leftarrow \|$NCaC_pos$[\mathbf{e}][..., 0]$ - NCaC_pos$[..., 2]\|^2$
10:    distances $\leftarrow$ [CatoNCaC_distances, NtoC_distances, CtoN_distances]
11:    $\mathbf{n} \leftarrow$ sinusoidal_positional_encoding(distances, dim=20, freq=1/50)
12:    $\mathbf{n} \leftarrow$ reshape($\mathbf{n}$, shape=(batch_size, 20 * 5 * 20))
13:    $\mathbf{n} \leftarrow$ linear_layer($\mathbf{n}$, output_dim=64)
14:    $\mathbf{f} \leftarrow [\mathbf{f}, \mathbf{n}]$
15:    return $\mathbf{f}$

---

**Algorithm 16** Fully Connected Residual Block

---

1: **function** FC_RES_BLOCK($\mathbf{f}$, output_dim)
2:     $\mathbf{y} \leftarrow$ linear_layer($\mathbf{f}$, output_dim)
3:     $\mathbf{f} \leftarrow \mathbf{f} + \mathbf{y}/\sqrt{2}$
4:     $\mathbf{f} \leftarrow$ relu($\mathbf{f}$)
5:     $\mathbf{f} \leftarrow$ layer_norm($\mathbf{f}$)
6:     return $\mathbf{f}$

---

---

**Algorithm 17** Backbone Frame Module

---

1: **function** BACKBONE_FRAME_MODULE($\mathbf{f}$, $\mathbf{F}$)
2:     $\mathbf{f} \leftarrow$ fc_res_block($\mathbf{f}$, output_dim=256)
3:     $\mathbf{f} \leftarrow$ fc_res_block($\mathbf{f}$, output_dim=256)
4:     $\mathbf{f} \leftarrow$ fc_res_block($\mathbf{f}$, output_dim=256)
5:     NCaC_shift $\leftarrow$ linear_layer($\mathbf{f}$, output_dim=9)
6:     NCaC_shift $\leftarrow$ reshape(NCaC_shift, shape=(batch_size, 3, 3))
7:     NCaC_new $\leftarrow$ $\mathbf{F} \circ$ NCaC_shift
8:     $\mathbf{F}' \leftarrow$ affine_from_3_points(NCaC_new)        ▷ See algorithm 21 in AlphaFold2 sup.
9:     $\mathbf{F}' \leftarrow$ mirror_x_and_z_axis($\mathbf{F}'$)        ▷ Same convention in AlphaFold2
10:    $\mathbf{x} \leftarrow \mathbf{F}'[..., 1]$
11:    return $\mathbf{x}$, $\mathbf{F}'$

---

**Algorithm 18** Transition Layer

---

1: **function** TRANSITION_LAYER($\mathbf{f}$)
2:     $\mathbf{f} \leftarrow$ fc_res_block($\mathbf{f}$, output_dim=256)
3:     $\mathbf{f} \leftarrow$ fc_res_block($\mathbf{f}$, output_dim=256)
4:     $\mathbf{f} \leftarrow$ fc_res_block($\mathbf{f}$, output_dim=256)
5:     return $\mathbf{f}$

---

**Algorithm 19** Cryo Key Vectors

---

1: **function** GET_CRYO_KEY_VECTORS($\boldsymbol{R}$)
2:     $\mathbf{f} \leftarrow$ conv3d($\boldsymbol{R}$, in_channels=1, out_channels=256, kernel_size=(1,3,3))
3:     $\mathbf{f} \leftarrow$ reshape($\mathbf{f}$, (batch_size, num_neighbours, -1))
4:     $\mathbf{f} \leftarrow$ relu(layer_norm($\mathbf{f}$))
5:     $\mathbf{f} \leftarrow$ linear_layer($\mathbf{f}$, output_dim=attention_heads $\times$ 256)
6:     $\mathbf{f} \leftarrow$ dropout($\mathbf{f}$, probability=0.5)
7:     $\mathbf{f} \leftarrow$ reshape($\mathbf{f}$, (batch_size, num_neighbours, attention_heads, 256))
8:     return $\mathbf{f}$

---

**Algorithm 20** Cryo Query and Value Vectors

---

1: **function** GET_CRYO_QUERY_VALUE_VECTORS($\mathbf{f}$, $\mathbf{e}$)
2:     $\mathbf{q} \leftarrow$ linear_layer($\mathbf{f}$, output_dim=attention_heads $\times$ 256)
3:     $\mathbf{q} \leftarrow$ reshape($\mathbf{q}$, (batch_size, attention_heads=8, 256))
4:     $\mathbf{v} \leftarrow$ linear_layer($\mathbf{f}$, output_dim=8 $\times$ 256)
5:     $\mathbf{v} \leftarrow \mathbf{v}[\mathbf{e}]$
6:     $\mathbf{v} \leftarrow$ reshape($\mathbf{v}$, (batch_size, num_neighbours=20, attention_heads=8, 256))
7:     return $\mathbf{q}$, $\mathbf{v}$

---

**Algorithm 21** Cryo Point Features

---

**Require:** global_variables
1: attention_heads $\leftarrow$ 8
2: num_features $\leftarrow$ 256
3: **function** GET_CRYO_POINT_FEATURES($\boldsymbol{C}$)
4:     $\mathbf{f} \leftarrow$ bottleneck($\boldsymbol{C}$, input_channels=1, hidden_channels, num_features / 4, expansion=4, stride=2)
5:     $\mathbf{f} \leftarrow$ bottleneck($\mathbf{f}$, input_channels=num_features, hidden_channels=num_features / 4, expansion=4, stride=2)
6:     $\mathbf{f} \leftarrow$ bottleneck($\mathbf{f}$, input_channels=num_features, hidden_channels=num_features / 4, expansion=4, stride=2)
7:     $\mathbf{f} \leftarrow$ bottleneck($\mathbf{f}$, input_channels=num_features, hidden_channels=num_features / 4, expansion=4, stride=2)
8:     $\mathbf{f} =$ reshape(batch_size, num_features, x, y, z)
9:     $\mathbf{f} \leftarrow$ spatial_mean($\mathbf{f}$)
10:    $\mathbf{f} \leftarrow$ linear_layer($\mathbf{f}$, output_dim= attention_heads $\times$ num_features)
11:    $\mathbf{f} \leftarrow$ reshape(batch_size, attention_heads, num_features)
12:    return $\mathbf{f}$

---

## A.4 COMPUTATIONAL CHARACTERISTICS

### A.4.1 NUMBER OF PARAMETERS

Segmentation network: 61,158,147

GNN total: 192,090,873

- Spatial IPA module: 9,659,904 ($\times$ 8 layers)

- Cryo-EM module: 7,722,581 ($\times$ 8 layers)

- Sequence attention module: 6,499,348 ($\times$ 8 layers)

- Torsion angle predictor: 240,806

- Backbone update module: 200,457

- Local confidence predictor: 198,401

- Existence mask predictor: 198,401

- Transition layer: 198,144

### A.4.2 TRAINING TIME AND COMPUTE

The training was carried out on 8 NVIDIA A100 GPUs. We trained the segmentation network for 400,000 steps with a batch size of 16. Each step took 0.99 seconds on average, for a total training time of 4.6 days. We trained the GNN for 300,000 steps with a batch size of 16. Each step took 1.73 seconds on average, for a total training time of 6 days.

### A.4.3 INFERENCE TIME AND COMPUTE

Inference time depends on the size of the structure and its sequence length. Below, we report the total time (in minutes) that it took to run inference on a single NVIDIA A100 GPU for all structures in the test set.

| | | | |
|---|---|---|---|
| 7ode: 19.5 | 7oix: 7.0 | 7pt6: 104.2 | 7pt7: 88.9 |
| 7q1u: 8.8 | 7rzy: 9.9 | 7sba: 132.7 | 7sjn: 7.5 |
| 7szi: 4.2 | 7tu5: 8.3 | 7tvz: 14.2 | 7txv: 15.6 |
| 7uck: 283.5 | 7ugg: 9.1 | 7um0: 17.0 | 7unl: 5.0 |
| 7wug: 21.1 | 7xpx: 4.8 | 7px8: 11.4 | 7y9u: 5.2 |
| 7z1m: 58.0 | 7sr8: 5.8 | 8a2q: 17.8 | 8a3t: 105.3 |
| 8dtm: 7.5 | 7u50: 7.4 | 7w9l: 11.2 | 7yzk: 5.5 |

### A.5 SEQUENCE SIMILARITY CUTOFF DATASET

To test whether overfitting to structures that are similar to those used for training affected our results, we also analyzed the relative performance of ModelAngelo and DeepTracer on a new test set. For this, we chose 15 recent structures from the PDB that did not contain a single protein with a sequence similarity of more than 30% to any of the chains in the training set. As shown in figure A.5 and tables 5 and 6, the relative performance of DeepTracer and ModelAngelo does not change for this smaller, more unique test set. We note that structure 7e1e is only partially built in the PDB deposition, which impacts its $C^\alpha$ recall.

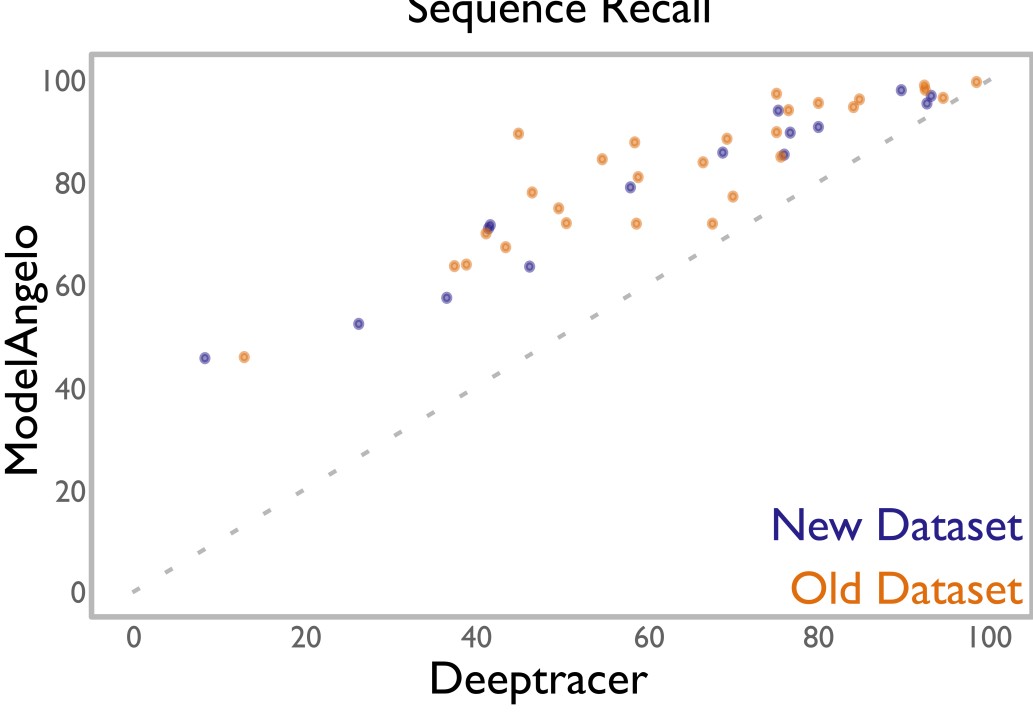

Figure 5: Scatter plot of results of the new sequence identity cutoff dataset, ModelAngelo pruned compared to Deeptracer. We note that the results are similar to the main dataset presented in the paper.

| PDB | Reso-lution | backbone RMSD | $C^\alpha$ recall | $C^\alpha$ precision | lDDT | Sequence match | Sequence recall |
|---|---|---|---|---|---|---|---|
| 8e2l | 3.51 | 0.663 | 59.8 | 97.3 | 90.6 | 96.0 | 57.4 |
| 7vt0 | 3.40 | 0.719 | 75.4 | 96.5 | 90.7 | 95.0 | 71.6 |
| 7uab | 3.70 | 0.610 | 54.8 | 98.5 | 92.6 | 95.4 | 52.3 |
| 7tnx | 3.50 | 0.577 | 94.6 | 98.9 | 93.2 | 99.3 | 94.0 |
| 7u58 | 3.10 | 0.484 | 87.3 | 96.7 | 95.1 | 97.9 | 85.4 |
| 8dze | 2.99 | 0.535 | 96.8 | 98.5 | 94.1 | 98.6 | 95.4 |
| 7e1e | 3.34 | 0.619 | 88.3 | 4.7 | 92.7 | 97.2 | 85.8 |
| 8aa5 | 2.46 | 0.410 | 73.6 | 99.5 | 97.1 | 96.4 | 71.0 |
| 7yer | 3.00 | 0.489 | 94.6 | 98.4 | 95.1 | 96.0 | 90.8 |
| 7ufg | 3.28 | 0.653 | 66.0 | 96.7 | 91.2 | 96.2 | 63.5 |
| 7uqx | 3.30 | 0.533 | 50.5 | 99.5 | 93.8 | 90.2 | 45.6 |
| 7uwq | 3.05 | 0.399 | 90.8 | 97.9 | 97.4 | 98.8 | 89.7 |
| 7uws | 3.47 | 0.599 | 80.9 | 79.0 | 91.7 | 97.6 | 79.0 |
| 7wjm | 3.30 | 0.426 | 98.0 | 99.6 | 96.6 | 98.9 | 96.9 |
| 7usc | 3.00 | 0.290 | 98.6 | 98.5 | 98.6 | 99.4 | 98.0 |

Table 5: ModelAngelo sequence similarity cutoff results

| PDB | Reso-lution | backbone RMSD | $C^\alpha$ recall | $C^\alpha$ precision | lDDT | Sequence match | Sequence recall |
|---|---|---|---|---|---|---|---|
| 8e2l | 3.51 | 1.404 | 66.9 | 93.7 | 82.7 | 54.5 | 36.5 |
| 7vt0 | 3.40 | 1.130 | 80.9 | 90.2 | 83.2 | 51.4 | 41.6 |
| 7uab | 3.70 | 1.231 | 83.1 | 95.7 | 84.8 | 31.5 | 26.2 |
| 7tnx | 3.50 | 0.975 | 95.2 | 88.0 | 89.0 | 79.1 | 75.3 |
| 7u58 | 3.10 | 0.807 | 85.7 | 96.8 | 91.8 | 88.7 | 76.0 |
| 8dze | 2.99 | 0.719 | 96.4 | 97.5 | 92.0 | 96.2 | 92.7 |
| 7e1e | 3.34 | 1.324 | 96.2 | 4.9 | 86.6 | 71.6 | 68.8 |
| 8aa5 | 2.46 | 1.010 | 72.8 | 87.8 | 91.1 | 56.8 | 41.4 |
| 7yer | 3.00 | 0.795 | 91.0 | 98.3 | 91.8 | 87.9 | 80.0 |
| 7ufg | 3.28 | 1.193 | 71.8 | 96.3 | 85.3 | 64.3 | 46.2 |
| 7uqx | 3.30 | 1.505 | 75.5 | 96.9 | 82.9 | 10.9 | 8.2 |
| 7uwq | 3.05 | 0.904 | 91.8 | 98.8 | 93.0 | 83.6 | 76.7 |
| 7uws | 3.47 | 1.157 | 87.9 | 72.9 | 85.4 | 66.0 | 58.0 |
| 7wjm | 3.30 | 0.685 | 97.6 | 99.6 | 92.7 | 95.6 | 93.2 |
| 7usc | 3.00 | 0.580 | 95.5 | 99.3 | 96.6 | 94.0 | 89.7 |

Table 6: Deeptracer sequence similarity cutoff results

A.6    EXPANDED MODEL COMPARISON

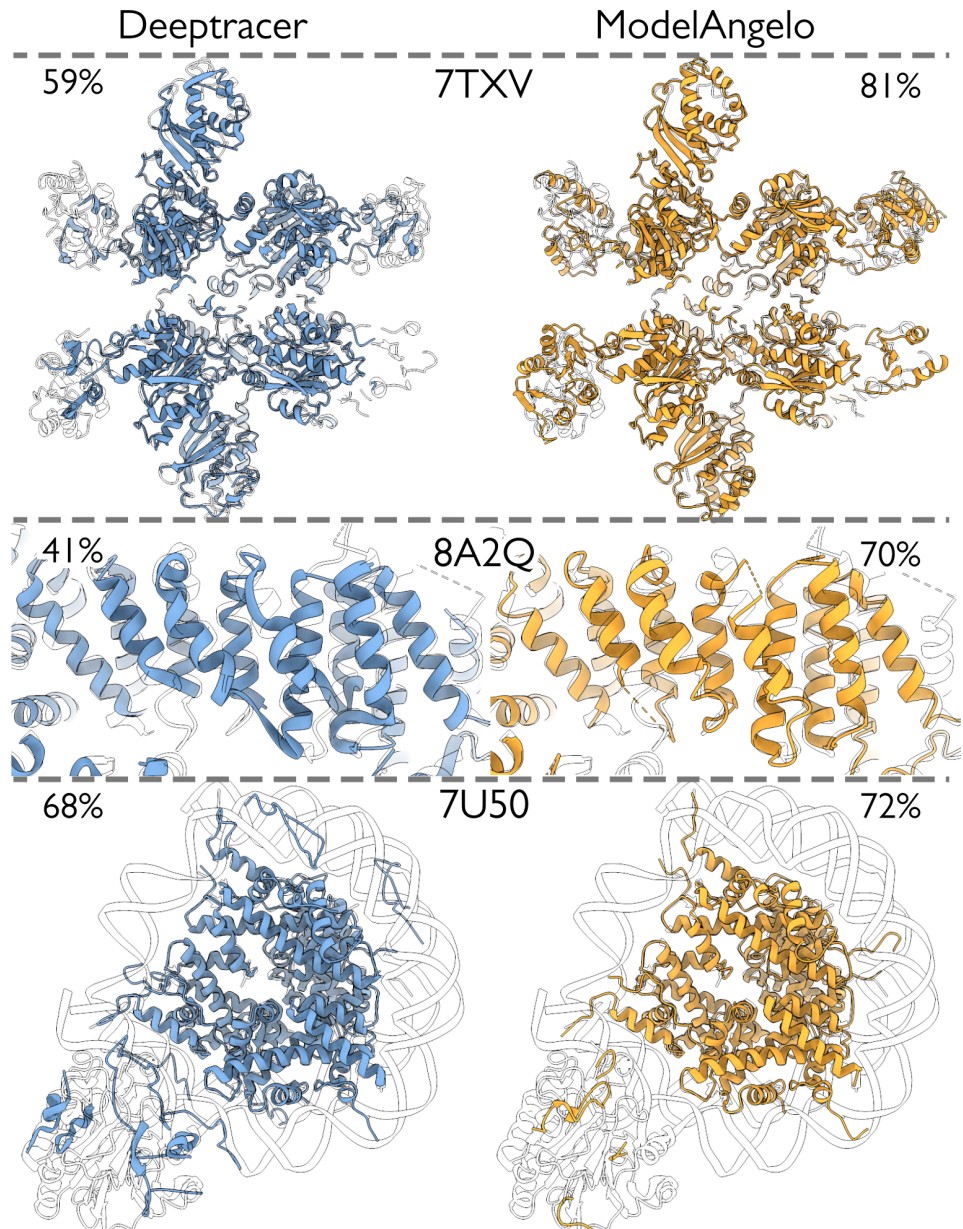

Figure 6: Additional results showing comparisons between Deeptracer (left) and ModelAngelo (right). Both models are viwed in color over the outline of the ground truth (deposited) model. The percentages show respective sequence recall for each model. Close-up on 7SJN highlights the incorrect chain connections made by Deeptracer. A global view of the models for 7TXV and 7U50 are shown.

