# OpenReview forum: "A Graph Neural Network Approach to Automated Model Building in Cryo-EM Maps"
_ICLR.cc/2023/Conference — ICLR 2023 poster_

### Official Review · Reviewer_9Yaz · 2022-10-21

**Confidence:** 2
**Correctness:** 3
**Technical Novelty And Significance:** 3
**Empirical Novelty And Significance:** 3
**Recommendation:** 8

**Clarity, Quality, Novelty And Reproducibility:**

The paper involves a lot of specialized knowledge and terminologies. The overall paper is difficult for me to read.  The machine learning part is mostly clear, but lacks a formal definition of the entire problem and some sub-problems (3.2 and 3.3).
The work is novel and of good quality in terms of a deep learning based cryo-EM analysis tool.

**Strength And Weaknesses:**

Strength
    1. Automated model building in cryo-EM maps is an important problem worth studying. ModelAngelo is among the initiated works that use deep learning techniques.
    2. Deep learning approaches used in ModelAngelo are technically sound. Some improvements in model architectures and learning objectives are successfully applied.
    3. The performance of ModelAngelo is empirically verified by a test dataset.

Weaknesses
    1. A mathematical formulation for the entire problem is missed. Though the problem is complex and difficult to be solved with an end-to-end framework, the original formulation is still needed at the beginning of the Methods section, followed by a brief introduction of the entire framework, i.e. how to split the entire task into several components and what specific role does each component play. I also suggest authors present a figure to illustrate the overall computing procedure of ModelAngelo.
    2. Quantitive evaluation results in Figure 3 only reflect middle outputs rather than the final outputs. Figure 4 illustrates the comparison of final results with a single data sample. Thereby, current evaluations are not convincing enough to confirm ModelAngelo’s superiority to competitors. Is it possible for a quantitive comparison on the final outputs?
    3. Many losses (e.g. (RMSD) loss, backbone RMSD loss, amino-acid classification loss, local confidence score loss, torsion angles loss, and full atom loss) are involved in the learning of GNN. What are the definition of these losses? How do you get the ground truth labels required (if so) for these losses?

**Summary Of The Paper:**

This paper introduces a machine learning tool ModelAngelo for automated model building in cryo-EM maps. The tool consists of a sequence of three components, including an improved Residual network for residue segmentation, a GNN-based framework for refining and an HMM for the final-mapping. These components are separately trained in a supervised fashion. ModelAngelo is evaluated in a test dataset of 28 map-model pairs, and shows better performance in sequence recall and backbone RMSD. The authors also discuss limitations and future research of the current implementation.

**Summary Of The Review:**

The paper introduces ModelAngelo for automated model building in cryo-EM maps. Compared with previous work (deeptracer), ModelAngelo largely improved deep learning techniques. It achieves better results at the cost of affordable increased complexity. Overall, I think this work is useful.

---

> ### Author Response · Authors · 2022-11-11
> **Formal definitions and clearer text**
>
> > 1. A mathematical formulation for the entire problem is missed. Though the problem is complex and difficult to be solved with an end-to-end framework, the original formulation is still needed at the beginning of the Methods section, followed by a brief introduction of the entire framework, i.e. how to split the entire task into several components and what specific role does each component play. I also suggest authors present a figure to illustrate the overall computing procedure of ModelAngelo.
>
> The manuscript now focuses more on the most novel aspect of our work: the GNN. Figure 2A describes its overall computing procedure of this network. Details are given in the pseudo-code in Appendix B. We have also now added mathematical formulations for both the graph initialization and graph refinement parts of the program, as the reviewer mentioned. Please also see [general comment (1)](https://openreview.net/forum?id=65XDF_nwI61&noteId=nXbP9pNd3G)
>
> > 2. Quantitive evaluation results in Figure 3 only reflect middle outputs rather than the final outputs. Figure 4 illustrates the comparison of final results with a single data sample. Thereby, current evaluations are not convincing enough to confirm ModelAngelo’s superiority to competitors. Is it possible for a quantitive comparison on the final outputs?
>
> Figure 3 presents the final output of the GNN after post-processing. This has now been clarified in the figure caption. Sequence recall of the final output accounts for both missed detections of Cα atoms in the segmentation task, errors in the residue type identification and the final mapping of the sequence to the backbone. Hence we believe it to be a good description of the quality and usefulness of the final model. Furthermore, it is known in other papers as “completeness of the built model” and is used widely as the definite metric for comparisons.
>
> > 3. Many losses (e.g. (RMSD) loss, backbone RMSD loss, amino-acid classification loss, local confidence score loss, torsion angles loss, and full atom loss) are involved in the learning of GNN. What are the definition of these losses? How do you get the ground truth labels required (if so) for these losses?
>
> These losses are now explicitly mentioned in the Methods section and they are all defined in Appendix A.1. The ground truth values are all calculated from atom coordinates in the training data. This should now be clear in the Appendix section for each loss.
>
> > The paper involves a lot of specialized knowledge and terminologies. The overall paper is difficult for me to read. The machine learning part is mostly clear, but lacks a formal definition of the entire problem and some sub-problems (3.2 and 3.3). The work is novel and of good quality in terms of a deep learning based cryo-EM analysis tool.
>
> We have added more formal definitions of the sub-problems, and made an overall attempt at increasing the readability of the paper. Please also see [general comment (1)](https://openreview.net/forum?id=65XDF_nwI61&noteId=nXbP9pNd3G).
>
> **We appreciate that the reviewer recognizes the novelty in the method and improvements in the results. We would also like to thank the reviewer, who’s comments have helped improve the quality of the manuscript. We hope that our major revision would be grounds for raising their score.**

---

> > ### Comment · Reviewer_9Yaz · 2022-11-28
> > **Response to authors**
> >
> > The authors’ response and updated manuscript addressed most of my concerns. Therefore, I raise my score from weak accept to accept.

---

> ### Author Response · Authors · 2022-11-16
> **Friendly reminder for rebuttal**
>
> This is a friendly reminder that we would very much value your input on our revised manuscript. Your comments have been extremely helpful in improving our paper. To aid you in assessing these changes, we have uploaded a tracked-changes PDF of our revision: see [general comment (3)](https://openreview.net/forum?id=65XDF_nwI61&noteId=nXbP9pNd3G). We would appreciate your views on any remaining points of contention, so we could still address those before the deadline (which is in *3 days*).

---

### Official Review · Reviewer_7rzk · 2022-10-26

**Confidence:** 3
**Correctness:** 4
**Technical Novelty And Significance:** 4
**Empirical Novelty And Significance:** Not applicable
**Recommendation:** 6

**Clarity, Quality, Novelty And Reproducibility:**

In general, the authors provide a good explanation of the importance of the task and its complexity. They also report promising results. However, the paper lacks many details, crucial for readers to understand the presented method and its training procedure. Given only the information provided in this manuscript, I have serious doubt the proposed method can be fully understood, let alone be reproduced.

**Strength And Weaknesses:**

Strength:
1. The description of the problem in question is sufficiently thorough.
2. The authors argue that the proposed method allows to build an atomic model de novo in the matter of hours compared to weeks in case of doing it manually.
3. ModelAngelo outperforms current state-of-the-art DeepTracer in terms of sequence recall and RMSD, although it’s slower due to increased overall computational needs. (Personally I don’t know if there are other baselines the authors could/should compare against.)
4. The proposed system can also provide confidence of its own predictions thanks to a special loss term.
5. Contrary to DeepTracer, the proposed method outputs side-chain coordinates.
6. Authors provide extensive analysis of limitation of the proposed method and possible directions of future work.

Weaknesses:
1. No schematic description of the model architecture and training procedure. This made it hard for me to be sure what is really proposed and to check if I understood the overall setup of the proposed method (which certainly has many moving parts).
2. Overall, the proposed algorithm is rather vaguely described.
3. The same thing is also true for the proposed loss, which I would love to see better on more thoroughly described.
4. I do know some things about cryo-EM data and workflows, but still I would have appreciated a better description of the data that goes into the proposed model.
5. Hard to interpret and somewhat limited description of the obtained results.
6. No ablation studies are presented, leaving many canonical questions on how important different parts of the proposed method are completely unaddressed.

**Summary Of The Paper:**

The paper proposes a method for the automated building 3D protein models from cryo-EM data. The proposed model combines a CNN with a GNN component in an attention-based architectures and also requires a HMM based post-processing step.

As a first step, the authors try to approximate the location of residues by solving a segmentation task. The model is a 3D FPN trained on Cryo-EM maps in a supervised fashion, using a customized weighted loss to address data imbalance problems. Training starts using a small, manually curated subset of the data. Results are later used to prune larger datasets.
Subsequently, the resulting map is processed by 3-stage GNN, which aims at correcting the CNN output. First module of the GNN is Cryo-EM Attention, which uses local neighborhood information by utilizing a novel graph-attention. Second is a Sequence Attention module, which is a BERT-like transformer taking as input the embedding computed by a pre-trained language model. The third and final module is Spatial Invariant Point Attention (IPA), inspired by similarly named module from AlphaFold2. The whole model is trained sequentially using several loss terms and produces 2 types of output. In the unpruned output, predicted chains are almost as-is from the output of the model, with minimal post-processing. This allows for or a larger percentage of the model to be built and gives higher recall. However, this may produce overhead for users in form of manually pruning longer sequences.
Therefore, the authors provide an additional post-processing mechanism based on hidden Markov models, which produces lower recall results, but higher sequence matching percentages, Authors argue that this is better for end users.

**Summary Of The Review:**

The paper proposes a multi-faceted method for solving an important and challenging task. The method contains several technical novelties but fails to fully clarify their details. The problem is described in sufficient detail, but several important issues with the way material is presented make it hard to completely understand the details of the implementation.

---

> ### Author Response · Authors · 2022-11-11
> **Improved manuscript and ablation studies**
>
> > ModelAngelo outperforms current state-of-the-art DeepTracer in terms of sequence recall and RMSD, although it’s slower due to increased overall computational needs. (Personally I don’t know if there are other baselines the authors could/should compare against.)
>
> We are now more explicit about the lack of useful automated model building tools for cryo-EM maps in the Introduction. To further illustrate this, we now also report the performance of Phenix (see Appendix section A.3, table 4), which is markedly worse than both DeepTracer and our approach.
>
> >No schematic description of the model architecture and training procedure. This made it hard for me to be sure what is really proposed and to check if I understood the overall setup of the proposed method (which certainly has many moving parts). [...] Overall, the proposed algorithm is rather vaguely described.
>
> We agree with the reviewer that the clarity of the paper needed improvement. We have now extensively rewritten the methods section with this in mind. Although we have not added a figure to further describe the model architecture, we have added technical details and a formal definition of the GNN. This includes a detailed algorithm description in the appendix and a formal description in the method section. Please also see [general comment (1)](https://openreview.net/forum?id=65XDF_nwI61&noteId=nXbP9pNd3G).
>
> > The same thing is also true for the proposed loss, which I would love to see better on more thoroughly described.
>
> The Methods section now explicitly mentions the losses, which are all defined in Appendix A.1.
>
> > I do know some things about cryo-EM data and workflows, but still I would have appreciated a better description of the data that goes into the proposed model.
>
> We agree with the reviewer’s criticism, as this was previously vaguely described. To wit, the Methods section now describes all inputs and output of the graph initialization and the graph refinement steps. Please also see [general comment (1)](https://openreview.net/forum?id=65XDF_nwI61&noteId=nXbP9pNd3G) for more detail about how we have strived to improve the clarity of the paper.
>
> > Hard to interpret and somewhat limited description of the obtained results. [...] No ablation studies are presented, leaving many canonical questions on how important different parts of the proposed method are completely unaddressed.
>
> We agree with the reviewer about the need for further clarity of the results section and inclusion of ablation studies to answer questions about the relative importance of different parts of the method. Therefore, we have now rewritten the results section, ran ablation experiments, and included these results in the Result section.
>
> **We would like to thank the reviewer for recognizing the novelty and superior results of ModelAngelo. We have addressed their excellent points about the clarity of the paper, request for additional comparisons, descriptions of the losses, and request for ablation studies. We hope that this could be a reason to raise their score.**

---

> > ### Comment · Reviewer_7rzk · 2022-11-30
> > **Thanks for your improvements.**
> >
> > I agree, the manuscript is much improved. It is still not an easy ready (mainly for people who know little about cryoEM data and workflows), but the results are great and I now believe the manuscript should be published.
> > At the same time I'm much looking forward to a journal version so can make good use of the additional space in main text and supplement... ;)

---

> ### Author Response · Authors · 2022-11-16
> **Friendly reminder for rebuttal**
>
> This is a friendly reminder that we would very much value your input on our revised manuscript. Your comments have been extremely helpful in improving our paper. To aid you in assessing these changes, we have uploaded a tracked-changes PDF of our revision: see [general comment (3)](https://openreview.net/forum?id=65XDF_nwI61&noteId=nXbP9pNd3G). We would appreciate your views on any remaining points of contention, so we could still address those before the deadline (which is in *3 days*).

---

### Official Review · Reviewer_RqeM · 2022-10-31

**Confidence:** 4
**Correctness:** 3
**Technical Novelty And Significance:** 2
**Empirical Novelty And Significance:** 2
**Recommendation:** 5

**Clarity, Quality, Novelty And Reproducibility:**

**Clarity** : Not clear enough. As I mentioned in above weakness point 1.

**Quality** : Fair but not enough.

**Novelty** : Limited as I mentioned in above weakness point 4.

**Reproducibility** : Hard to reproduce, no architecture details provided.


**Strength And Weaknesses:**

## Strengths
1. They propose to use protein sequences as input to better the structure prediction results, which is different from previous work named Deeptracer.
2. They employ a GNN to integrate information from different domains including Cryo-em map, protein sequence, and alpha carbon positions.

## Weaknesses:
1. This paper is not well-written and hard to follow. The problem definitions of these two steps are unclear. (1) Clear definitions of inputs, outputs, and data structures are missing. It is hard to guess how they use these different features from different domains and what information is exactly used for the second step. (2) Some operations are hard to follow. How can they use literature bonds to update the coordinates?
2. The experimental part is not convincing. The comparison with Deeptracer is problematic because (1) Deeptracer and ModelAngelo use different training data, which means this is not a fair comparison, and (2) compared with Deeptracer, additional protein sequence information is used as input for ModelAngelo, which is a huge assumption to the problem setting.
3. Ablation studies are missing. The positions of alpha carbons can be obtained by the first stage only. Ablation studies are needed for a better understanding of the whole framework.
4. The proposed method is not novel enough. The methods for the first stage and the second stage are borrowed from previous papers with some modifications to better fit the problem settings, with limited novelty for the model part.


**Summary Of The Paper:**

This paper proposes ModelAngelo, a two-step pipeline to predict the protein structure using cryo-EM maps and protein sequences as input. To do this, ModelAngelo first employs a 3D CNN to predict positions of alpha carbons in protein sequences taking cryo-EM maps as input. Apart from this, they train a separate graph neural network to finish the task of position de-noising by using data augmentation methods, which can be used to fine-tune the protein structure after training. These two steps are separately trained. The employed models of CNN and GNN are task-driven but not new.

**Summary Of The Review:**

Overall, because of the major concerns listed in the above weaknesses, I vote for the rejection of this paper.

---

> ### Author Response · Authors · 2022-11-11
> **Rewritten methods and formal descriptions**
>
> > This paper is not well-written and hard to follow. The problem definitions of these two steps are unclear. (1) Clear definitions of inputs, outputs, and data structures are missing. It is hard to guess how they use these different features from different domains and what information is exactly used for the second step. (2) Some operations are hard to follow. How can they use literature bonds to update the coordinates?
>
> We agree with this criticism, which has also been mentioned by other reviewers. Therefore, we have expanded the methods section to include more technical details and a formal description about the GNN as a whole and the different modules. All operations are clarified in the algorithm sections of the appendix (see A.4). The projection to literature bonds for the backbone atoms has now been clarified at the end of section 3.2.1. Please also see [general comment (1)](https://openreview.net/forum?id=65XDF_nwI61&noteId=nXbP9pNd3G)
>
> > The experimental part is not convincing. The comparison with Deeptracer is problematic because (1) Deeptracer and ModelAngelo use different training data, which means this is not a fair comparison, and (2) compared with Deeptracer, additional protein sequence information is used as input for ModelAngelo, which is a huge assumption to the problem setting.
>
> DeepTracer is current state-of-the-art. We have now also compared our approach to the Phenix software, which has much worse results than our approach or DeepTracer. Our new ablation studies show that combining information from all three modules of our approach injects useful information into the problem, and we demonstrate that our method produces better models than DeepTracer. Please note that Deeptracer *also* requires the sequence as an input for post-processing, without which the results are much worse. We merely make better use of these data by having it as an input to our method as well.
>
> We also respectfully disagree with the reviewer that “additional protein sequence information is a huge assumption in the protein setting”. Cryo-EM is mostly performed on proteins that are recombinantly expressed in the lab. This requires all protein sequences to be known. Even in cases where proteins purified from native sources are used in a cryo-EM study, most protein sequences will typically be known from the now widely available genomes of model organisms. We discuss the (relatively uncommon) case of unknown sequences in the Discussion section.
>
>
> > Ablation studies are missing. The positions of alpha carbons can be obtained by the first stage only. Ablation studies are needed for a better understanding of the whole framework.
>
> The reviewer is correct that the alpha carbon positions can be obtained by the graph initialization stage, but that is not sufficient for full atomic model building, which also requires the amino-acid assignments, the backbone affine frames and the side-chain torsion angles, which are only calculated in the GNN. The Methods section now explains this better. We have also added ablation studies for the modules used in the GNN to the Results section.
>
> > The proposed method is not novel enough. The methods for the first stage and the second stage are borrowed from previous papers with some modifications to better fit the problem settings, with limited novelty for the model part.
>
> We have focused the revised manuscript on the second step: the GNN. This step is novel in that our work represents the first time that all three sources of information (cryo-EM map, sequence and graph topology) have been combined in a single approach for automated model building in cryo-EM maps. Experimental structure determination by cryo-EM is a fundamentally different problem setting from structure prediction. Existing state-of-the art methods for automating atomic modelling in cryo-EM maps were not yet a viable alternative to manual building, but our approach is likely to change that.
>
> **We thank the reviewer for their insightful comments that have helped improve the overall quality of the manuscript. We have extensively rewritten the method section to clarify the technical description and the novel aspect of our work. We have added a more formal description also on the GNN along with an extension on the formal definition of the segmentation model. We have also added more descriptions of the algorithms in the Appendix, to aid in clarity and reproducibility. In light of this, we hope the reviewer would consider raising their score.**

---

> > ### Comment · Reviewer_RqeM · 2022-11-11
> > **Thank you for your response**
> >
> > Could you please mark your edited parts as red such that I can see the differences? After re-checking, if some of the concerns are addressed, I will consider raising the score.

---

> > > ### Author Response · Authors · 2022-11-11
> > > **Highlighted differences in the revision**
> > >
> > > We have now uploaded an automatically generated report that highlights the changes made to the main text. It can be found in the [supplementary files](https://openreview.net/attachment?id=65XDF_nwI61&name=supplementary_material). In this report, text insertions are marked with blue, replacements are marked with orange and deletions are marked with red.
> > >
> > > Please also note that sections A.3 to A.6 have been added to the appendix in this revision, which is not shown in the above mentioned report file.

---

> > > > ### Comment · Reviewer_RqeM · 2022-11-14
> > > > **Response to authors**
> > > >
> > > > The concern about weakness 1 is addressed. The clarity of the method is improved, but I still can not see how you update the coordinates using bond lengths in the main paper. Sorry if I miss anything.
> > > >
> > > > The concern about ablation study is addressed with the updated figure in the main paper.
> > > >
> > > > For the novelty of the proposed method, I still feel that there is not so much novelty beyond the additional inputs of other modalities to the graph neural network.
> > > >
> > > > Thank you for your response and I increased my score.

---

### Official Review · Reviewer_AJLC · 2022-11-03

**Confidence:** 4
**Correctness:** 3
**Technical Novelty And Significance:** 3
**Empirical Novelty And Significance:** 4
**Recommendation:** 8

**Clarity, Quality, Novelty And Reproducibility:**

The quality and novelty of the work is high. The clarity of the paper could be improved.

**Strength And Weaknesses:**

Strengths include novelty and good empirical performance. This work is to my knowledge the first example of integrating learning over protein sequences to the task of atomic model fitting into cryo-EM maps. The algorithm is designed to iteratively incorporate information from the cryo-EM volume, geometric considerations from modeling the protein chain, and the underlying sequence.

One weakness is that the writing could be improved. The writing is clear and explains the design of the method well, however, some statements tend to be anecdotal and not substantiated. e.g. It would be nice to show ablations of the different model components. The details of the algorithms are also missing.

Questions:
It was not clear to me how/when side chain atoms are fit
The postprocessing step is also a bit unclear. How is the chain inferred? Why is the HMMER sequence search needed if the input sequence is known?
How does the model deal with multiple sequences in a large protein complex?

**Summary Of The Paper:**

This work leverages recent advances in large-scale biological machine learning to the task of atomic model fitting into cryo-EM density maps. There are three stages of this pipeline: 1) a residue segmentation model that identifies C_alpha locations in the cryo-EM density map, 2) a GNN-based model which refines the C_alpha locations by sequentially attending over the cryo-EM volume, the underlying protein sequence, and the predicted C_alpha locations, and 3) a postprocessing step where HMMER is used to infer the sequence and relax the geometry of the final atomic model. The model was applied on 28 test cryo-EM maps and compared against DeepTracer.


**Summary Of The Review:**

This is fantastic work that should be accepted at ICLR. It is a novel application of machine learning to the manual, labor-intensive task of atomic model fitting in cryo-EM maps. Empirical results are strong, and the algorithms are well-designed. The writing of the paper could however be improved.

---

> ### Author Response · Authors · 2022-11-11
> **Improved manuscript and more details**
>
> > One weakness is that the writing could be improved. The writing is clear and explains the design of the method well, however, some statements tend to be anecdotal and not substantiated. e.g. It would be nice to show ablations of the different model components. The details of the algorithms are also missing.
>
> We agree with this criticism, which has also been mentioned by other reviewers. Therefore, we have expanded the method section to include more technical details, a formal definition of the GNN and clarified the existing text. We have also added ablation studies. Please also see [general comment (1) and (2)](https://openreview.net/forum?id=65XDF_nwI61&noteId=nXbP9pNd3G).
>
> > It was not clear to me how/when side chain atoms are fit The postprocessing step is also a bit unclear. How is the chain inferred? Why is the HMMER sequence search needed if the input sequence is known? How does the model deal with multiple sequences in a large protein complex?
>
> An explicit description of how the main chain and side chain coordinates are determined is now included in the Methods section: they are constructed from affine frames for the main chain and predicted torsion angles for the side chains, using methods that are inspired by AlphaFold2. More details are available in the pseudo-code in Appendix A.4. We have also added a section to explain why the HMMER sequence search is still needed, see second paragraph in section 3.3.
>
> **We thank the reviewer for pointing out the shortcomings and making suggestions, which we believe have helped improve the quality of the text. We would be thrilled if the reviewer would deem these changes worthy of a higher score.**

---

> ### Author Response · Authors · 2022-11-16
> **Friendly reminder for rebuttal**
>
> This is a friendly reminder that we would very much value your input on our revised manuscript. Your comments have been extremely helpful in improving our paper. To aid you in assessing these changes, we have uploaded a tracked-changes PDF of our revision: see [general comment (3)](https://openreview.net/forum?id=65XDF_nwI61&noteId=nXbP9pNd3G). We would appreciate your views on any remaining points of contention, so we could still address those before the deadline (which is in *3 days*).

---

### Official Review · Reviewer_K3kR · 2022-11-04

**Confidence:** 4
**Correctness:** 3
**Technical Novelty And Significance:** 4
**Empirical Novelty And Significance:** 4
**Recommendation:** 5

**Clarity, Quality, Novelty And Reproducibility:**

The method is novel and the results appear to be good. The paper is clear but far too high level for an ML venue. It would not be possible to implement this model from the paper, because insufficient details are provided. The evaluation also does not sufficiently explore and justify some of the modeling decisions.

**Strength And Weaknesses:**

Overall, the paper shows promising results. I am, however, concerned by the way the method is presented for a machine learning audience. The method section is vague and does not provide sufficient detail to be able to fully understand and reproduce the method. There is also a lack of detailed descriptions about the neural network architecture and ablation studies. The paper would benefit from further discussion, detailed explanation, and clarification.

1. Other works combining sequence with cryo-EM: The authors mentioned several other works utilizing protein structure prediction programs to improve output predictions to fit the cryo-EM map (page 3). What are the main differences between those works and ModelAngelo? The authors claimed that “Such approaches are likely to propagate errors in the structure prediction. ”. What is the basis of this claim and what makes ModelAngelo not susceptible to the same problems?

2. Methods section 3.1 describes the coarse location estimation module, but there are many important details left to the readers imagination. How is the dataset split into train, test and validation sets? How is the neural network evaluated and validated? What is the performance of this network? What are the choices of layers for this CNN and what is the reason behind these choices? Are there any ablation studies done for the design?

3. Methods section 3.2: Again, the description of the GNN is vague. There is no description about the training, testing and validation processes nor evaluation of the model performance. Specifically how is this module designed?

4. For eq(1),  χ is the characteristic function of whether x contains a Cα atom. How is this function defined?

5. Time and computing resources are briefly discussed, but there does not seem to be information regarding training time for each part of the method nor the computing resources used. What kind of hardware is needed to train and generate predictions from ModelAngelo?

6. The test dataset is small and the similarity between the train and test structures is unclear. It would be helpful to see evaluation of ModelAngelo on a larger number of structures and to stratify performance based on similarities to the proteins in the training set. Is it possible that some of the test proteins are structurally similar to proteins in the training set? How well does ModelAngelo perform on structures substantially dissimilar to those seen during training? Structural splits based on structural alignment scores or SCOP/CATH classes would be illuminating.

7. How does ModelAngelo compare with more conventional approaches to model building such as Phenix or Rosetta?  we would like to see more metrics in comparison as well as comparison with more approaches such as Phenix and Rosetta. What about MAINMAST?

8. The description of each algorithm could be more detailed. For example, how do get_cryo_key_vectors and get_cryo_point_features work? What does get_query_vector_in_local_frame mean?


**Summary Of The Paper:**

This paper proposes a new method for automated model building of protein structures in cryo-EM maps. The method utilizes amino acid sequence data and prior knowledge about protein geometries in addition to cryo-EM data. Specifically, it starts with approximate locations of individual residues using a CNN and then uses a GNN that takes in information from sequence data, cryo-EM map and initial positions of atoms to refine the full atom positions of each residue. The main innovation is utilizing several aspects of protein data in contrast to traditional sequence only or cryo-EM only approaches. The results look promising, achieving better sequence recall and backbone root mean squared deviation compared to Deeptracer, though with only 28 evaluations and a longer execution time. The main limitations discussed are sensitivity to resolution, inability to achieve full atomic modeling of nucleic acids, and inability to construct models with unknown sequences.

**Summary Of The Review:**

The method is interesting and works well. However, the manuscript needs significant work to be better presented to the ML community. Model architecture and training details are vague and experiments do not demonstrate the value of individual model components. With some significant work on the text, I think this can be a strong contribution, but I think the manuscript as currently written is of limited value to the ML community.

What would raise my score: revise the manuscript to provide detailed architectural and training details and expand the evaluation to demonstrate the value of the components of the model and provide further characterization of the performance as it relates to structural similarity to proteins in the training set.

---

> ### Author Response · Authors · 2022-11-11
> **Improved manuscript and new data**
>
> > [...] There is also a lack of detailed descriptions about the neural network architecture and ablation studies. The paper would benefit from further discussion, detailed explanation, and clarification.
>
> In the current manuscript we have expanded on the technical aspects of the GNN, include a formal description, and added ablation studies for different modules of the GNN. Please also see [general comment (1) and (2)](https://openreview.net/forum?id=65XDF_nwI61&noteId=nXbP9pNd3G).
>
> > [...] The authors claimed that “Such approaches are likely to propagate errors in the structure prediction. ”. What is the basis of this claim and what makes ModelAngelo not susceptible to the same problems?
>
> We now mention that many proteins adopt highly distinct conformations in complex with other proteins and this is hard to predict using approaches like AlphaFold2. Incorrectly predicted folds of individual proteins may be hard to correct by flexible docking into cryo-EM maps. Our approach builds models “from the bottom up”, by combining individual residues that are placed in the cryo-EM map, which prevents the need for the flexible docking of larger molecules.
>
> > Methods section 3.1 describes the coarse location estimation module, but there are many important details left to the readers imagination. [...] Are there any ablation studies done for the design?
>
> See [general comment (1)](https://openreview.net/forum?id=65XDF_nwI61&noteId=nXbP9pNd3G). Pseudo-code describing this entire step is now given in Appendix A.4. The novelty of our approach lies mostly in the GNN, on which the paper now focuses more. Ablation experiments for the GNN network have now been done and included in the Results section. For ablation studies, please also see [general comment(2)](https://openreview.net/forum?id=65XDF_nwI61&noteId=nXbP9pNd3G).
>
> > Again, the description of the GNN is vague. [...]
>
> We have made extensive changes to the methods section of the paper to increase clarity. We have now also included pseudo-code for all important sections in Appendix A.4. Please also see comments above and [general comment (1)](https://openreview.net/forum?id=65XDF_nwI61&noteId=nXbP9pNd3G).
>
> > For eq(1), χ is the characteristic function of whether x contains a Cα atom. How is this function defined?
>
> The characteristic function χ is defined as whether the voxel x contains a Cα atom. This should now be clearer in the text.
>
> > Time and computing resources are briefly discussed, but there does not seem to be information regarding training time [...]
>
> We thank the reviewer for this point. We have now added a section on computational characteristics for training and inference, as well as timings for all 28 structures reported in the paper, in Appendix section A.4. Briefly, with 8 NVIDIA A100 GPUs, the training takes around a week each for the segmentation network and the GNN.
> > The test dataset is small and the similarity between the train and test structures is unclear. [...] How well does ModelAngelo perform on structures substantially dissimilar to those seen during training? [...]
>
> We thank the reviewer for this insightful question. Although structure similarity is a major problem for overfitting in protein structure prediction, we believe this to be less the case for our approach because each cryo-EM map already defines the overall fold of the protein. To test this, we created a second manually curated test set of 15 structures that do not have a single protein chain with a sequence similarity higher than 30% with any of the chains in our training set (as a proxy for structural similarity).
>
> The stringent selection criterion resulted in the selection of only relatively small structures. Nevertheless, the relative performance of our approach and DeepTracer remains similar as for the original test dataset, as we now show in Appendix section A.5. For the camera-ready version of this paper, we are planning on including a more comprehensive analysis of results based on the similarity of each chain’s sequence to a sequence that has been used in training.
>
> > How does ModelAngelo compare with more conventional approaches to model building such as Phenix or Rosetta? [...]
>
> We have now included a comparison with Phenix in Appendix section A.3, table 4. The results are much worse than both ModelAngelo and Deeptracer. In the time-span available for this rebuttal, we were unable to install and run Rosetta and MAINMAST.
>
> > The description of each algorithm could be more detailed. [...]
>
> We have rewritten the methods section with an emphasis on clarity. Also, pseudo-code for these algorithms is now provided in Appendix A.4.
>
> **We thank the reviewer for their insights and suggestions. We hope that the extensive changes to the manuscript and the inclusion of new data on ablation studies, Phenix and a more unique second test set, have addressed their concerns and would be grounds for raising their score.**

---

> ### Author Response · Authors · 2022-11-16
> **Friendly reminder for rebuttal**
>
> This is a friendly reminder that we would very much value your input on our revised manuscript. Your comments have been extremely helpful in improving our paper. To aid you in assessing these changes, we have uploaded a tracked-changes PDF of our revision: see [general comment (3)](https://openreview.net/forum?id=65XDF_nwI61&noteId=nXbP9pNd3G). We would appreciate your views on any remaining points of contention, so we could still address those before the deadline (which is in *3 days*).

---

### Author Response · Authors · 2022-11-11
**General Comments**

**General comments**

We thank all five reviewers for their insightful comments. Based on these, we have performed a major revision of the paper, which we feel has much improved over the initial submission. Below, we will first address the most important changes made in this revision, which were based on overlapping comments from multiple reviewers. We will then refer to these general comments when addressing each of the reviewer’s comments separately.


**(1) Paper clarity & reproducibility**

All reviewers agreed on a lack of clarity in our initial manuscript, in particular regarding the description of our approach. This lack of clarity also raised issues with the reproducibility of our work. We have taken this criticism to heart and made major revisions to the main text. The paper is now more focused on the graph neural network, which represents the main novelty of our work. Moreover, the Methods section now explicitly describes all inputs and outputs to the CNN that is used to initialise the graph, as well as the GNN that refines the graph. To further aid the understanding of our approach and to increase the reproducibility of our study, we now present detailed pseudo-code for the most important steps (see Algorithms 1-21 in Appendix A.3). We also note that our source code will be distributed under an open-source MIT license, and that we will add a reference to the relevant repository after de-anonymization of the paper.


**(2) Ablation studies**

Several reviewers asked for ablation studies to examine the relative effects of our cryo-EM, sequence and IPA modules. We have now performed these experiments (see Figure 3C). Without the cryo-EM module, our approach of model building in cryo-EM maps will not work. Ablation of either the sequence or the IPA module leads to worse results, in particular for maps at lower resolution (as reflected in higher atomic B-factors). Interestingly, when ablating both the sequence and the IPA module, our approach performs worse than DeepTracer, which also only uses the cryo-EM map. These results indicate that using all three modules together is indeed useful.


**(3) Highlighted differences in the revision**

Per the reviewer's request, several changes have been made to the manuscript. To simplify the reviewing process we have uploaded an automatically generated report that highlights the changes made to the main text. It can be found in the [supplementary files](https://openreview.net/attachment?id=65XDF_nwI61&name=supplementary_material). Text insertions are marked with blue, replacements are marked with orange and deletions are marked with red.

Please also note that sections A.3 to A.6 have been added to the appendix in this revision, which is not shown in the above mentioned report file.

---

### Decision · Program_Chairs · 2023-01-20

**Decision:**

Accept: poster

**Justification For Why Not Higher Score:**

While the whole process is an innovation, the method is a well-designed workflow based on existing ideas, which the authors applied to a key problem in a new way.

**Justification For Why Not Lower Score:**

The authors have extensively revised their work during the discussion and addressed most of the major concerns that several reviewers have.

**Metareview: Summary, Strengths And Weaknesses:**

In this work the authors proposes a new pipeline leveraging deep learning to automate the building of protein structural models from cryo-EM density maps. Their approach integrates information from protein sequence, EM density map, and graph topology to help improve the performance of model building and the authors tested and validated their method in a decent number of targets. There are still room for improvement in terms of complete clarity of the method details, more demonstration and discussion of the benchmarks. Overall this work developed a good and useful machine learning tool for an important real-world problem in protein structure biology.

While the initial reviews had a more positive tone, most reviewers were not too enthusiastic about this work due to several technical and presentation issues. It is commendable that the authors were able to improve significantly over the initial submission thanks to the reviewers' comments.

**Note From Pc:**

if the above contains the word "oral" or "spotlight" please see: "oral" presentation means -> notable-top-5% and "spotlight" means -> notable-top-25%. As stated in our emails, we are disassociating presentation type from AC recommendations